# TRAINING-FREE GROUP RELATIVE POLICY OPTIMIZATION

## ABSTRACT

Reinforcement Learning (RL) has emerged as a pivotal strategy for adapting Large Language Model (LLM) agents to specialized domains and complex tool-use scenarios. However, existing approaches typically instantiate the policy as a parameterized LLM, relying on gradient-based updates such as Group Relative Policy Optimization (GRPO). This paradigm incurs prohibitive computational costs and risks catastrophic forgetting, often making it impractical for resource-constrained scenarios. In this work, we propose a fundamental rethinking of agentic RL by introducing Training-Free Group Relative Policy Optimization (Training-Free GRPO). It instantiates the policy as a frozen LLM paired with a variable experiential context, shifting optimization from the parameter space to the context space. Mirroring the iterative structure of vanilla GRPO, our method replaces gradient descent with multi-epoch RL learning by introspecting on groups of trial-and-error rollouts, where the LLM extracts a *semantic group advantage* to iteratively refine its problem-solving experiences without parameter updates. Experiments on mathematical reasoning and web search tasks demonstrate that Training-Free GRPO establishes a new Pareto frontier between test-time performance and learning cost. Also, we show that applying our method to a frozen flagship LLM like DeepSeek-V3.1-Terminus using merely 100 training samples yields superior performance to fully fine-tuning a 32B LLM, while slashing learning costs by orders of magnitude from $800 to $8. It offers a highly effective and accessible pathway for optimizing LLM behaviors in real-world applications.

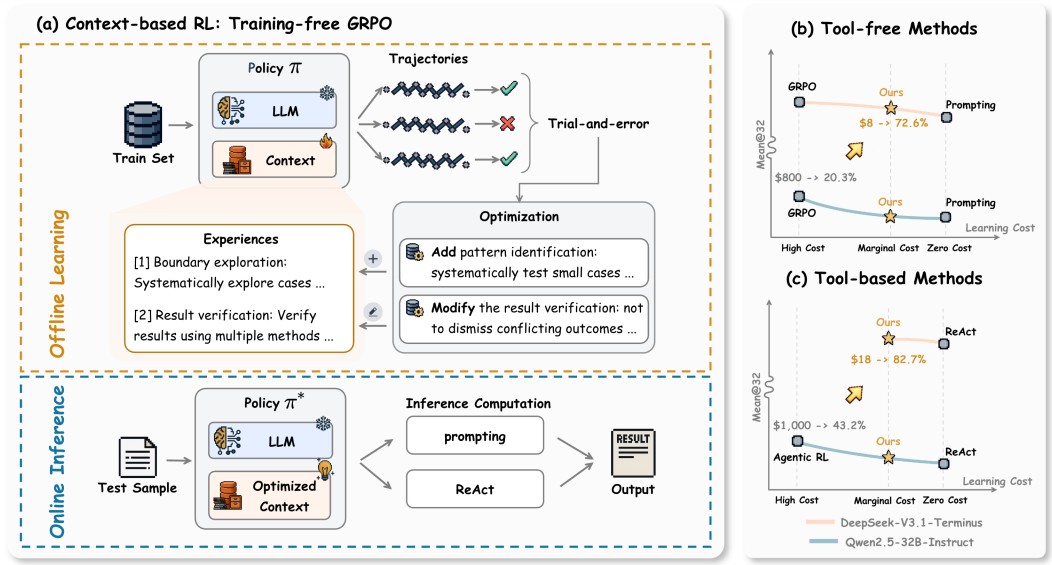

Figure 1: (a) Training-free GRPO instantiates RL policy as a frozen LLM paired with a variable experiential context, optimizing through trial-and-error process. After offline learning, such optimized experiences guide the LLM during online inference on test samples. (b-c) On AIME'24 benchmark, Training-free GRPO effectively interpolates the Pareto frontier with and without tool use, offering a significantly lower learning cost while still delivering meaningful performance improvements.

# 1 INTRODUCTION

Reinforcement Learning (RL) is a fundamental paradigm of learning through systematic trial-and-error (Kaelbling et al., 1996), where an agent interacts with an environment, observing states and executing actions to maximize reward signals by adjusting a *policy*. With the emerging capability of Large Language Models (LLMs) in complex, real-world environments (Mai et al., 2025; Xue et al., 2025; Jin et al., 2025; Team, 2025; Zhang et al., 2024; Huang et al., 2023; Wang et al., 2024b;a; Yuksekgonul et al., 2025), RL has become a pivotal strategy for adapting LLM agents to specialized domains and tools (Feng et al., 2025a; Tongyi DeepResearch Team, 2025; Tao et al., 2025; Li et al., 2025). Among these studies, the *policy* is typically instantiated as a parameterized LLM. And their policy optimization is usually based on gradient-based updates in the parameter space, employing Group Relative Policy Optimization (GRPO) (Shao et al., 2024) or its variants (Liu et al., 2025; Yu et al., 2025; Zheng et al., 2025). While these RL algorithms effectively enhance task-specific capabilities, their reliance on fine-tuning parameters poses significant practical challenges:

- **Computational Cost:** Even for smaller LLMs, fine-tuning demands substantial computational resources, making it both costly and environmentally unsustainable. For larger models, the costs become prohibitive.

- **Poor Generalization:** Parameters optimized for specific tasks often suffer from catastrophic forgetting, degrading cross-domain generalization. For practical applications with multiple subtasks, this necessitates deploying multiple specialized models which increases system complexity.

- **Data Scarcity:** Effective fine-tuning needs large volumes of high-quality annotated data that are scarce in specialized domains. With limited samples, LLMs are highly susceptible to overfitting.

- **Diminishing Returns:** In practice, resource constraints often result in fine-tuning smaller LLMs with fewer than 32 billion parameters. Paradoxically, larger API-based LLMs often deliver superior cost-effective service through deployment scalability, leading to the marginal gains of fine-tuning smaller models.

These limitations suggest a fundamental rethinking of RL in the LLM era. Indeed, RL extends far beyond gradient-based updates of parameterized models. By definition, a *policy* is simply a mapping from states to actions (Kaelbling et al., 1996). For example, the policy could be simple look-up tables (Gittins et al., 2011). Furthermore, even when neural networks serve as the policy, RL optimization can use gradient-free methods that search for optimal policies (Arulkumaran et al., 2017). This broader perspective leads to a critical question: *Given that a policy can be any mapping, and optimization is not strictly bound to gradient updates, must we incur the prohibitive cost and suffer the poor generalizability of updating LLM parameters in RL?*

Building upon this insight, we propose instantiating the RL *policy* as the union of a frozen LLM and its variable context, thereby shifting optimization from the parameter space to the context space. As illustrated in Figure 1(a), during multi-epoch RL process, the context evolves through trial-and-error. By leveraging the LLM's intrinsic In-Context Learning (ICL) capabilities (Brown et al., 2020), this strategy could achieve policy improvement without modifying a single model weight in RL. Specifically, we introduce **Training-Free Group Relative Policy Optimization (Training-Free GRPO)**, which mirrors the multi-epoch vanilla GRPO but replaces gradient descent with evolving experiences in the context. In each epoch, for every training sample, the agent generates a group of trials based on current experiences. Rather than calculating a numerical advantage for parameter tuning, LLM could introspect on these diverse outputs and distill a *semantic group advantage*, a textual optimization direction derived from contrasting successful and failed trials. Such advantage optimizes the current experiences in the context, serving as a refined policy for subsequent epochs.

By evaluating challenging mathematical reasoning and interactive web searching tasks, we demonstrate that Training-Free GRPO significantly enhances the performance of frozen LLMs, such as Qwen2.5-32B-Instruct (Yang et al., 2025a) and DeepSeek-V3.1-Terminus (DeepSeek-AI, 2024), using only dozens of training samples. As shown in Figure 1(b)-(c), it establishes a new Pareto frontier between test-time performance and learning costs, offering an effective and efficient alternative to both fine-tuning small LLMs and the direct usage of large LLMs.

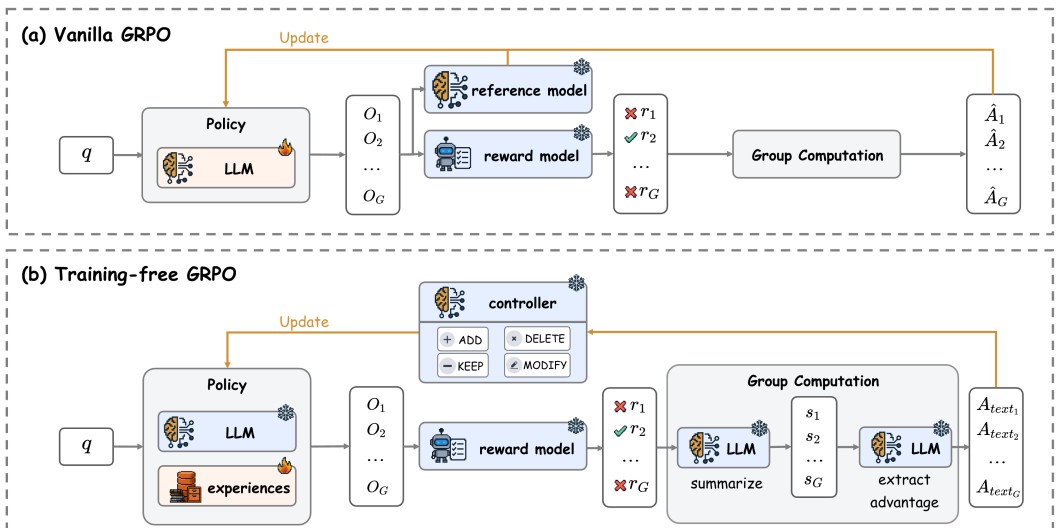

Figure 2: Comparison of vanilla GRPO and Training-Free GRPO.

Our principal contributions are summarized as follows:

- **A Context-based RL Paradigm:** We demonstrate that trial-and-error Reinforcement Learning can be effectively instantiated by utilizing a frozen LLM with evolving context as the policy, optimizing the context rather than model parameters.

- **Training-Free GRPO:** We propose the algorithm that computes *semantic group advantage* to iteratively refine the policy context, mirroring the vanilla GRPO process while eliminating costly gradient updates.

- **Efficiency and Generalization:** Training-Free GRPO achieves competitive performance with a fraction of the computational resources required for fine-tuning LLMs, better preserving the LLM's generalizability.

## 2 TRAINING-FREE GRPO

In this section, we introduce Training-Free GRPO that instantiates the RL *policy* as a frozen LLM with variable context, thereby achieving policy optimization in the context space without any LLM parameter update.

**Vanilla GRPO.** As shown in Figure 2, the vanilla GRPO instantiates the RL *policy* as a tunable LLM $\pi_\theta$. It operates by first generating a group of $G$ outputs $\{o_1, o_2, \ldots, o_G\}$ for a given query $q$ using the current policy LLM, i.e., $\pi_\theta(o_i \mid q)$. Each output $o_i$ is then independently scored with a reward model $\mathcal{R}$, which could be a rule-based function or an LLM judging whether $o_i$ matches the ground truth $y$, producing the scalar reward $r_i = \mathcal{R}(o_i, y)$. With rewards $\mathbf{r} = \{r_1, \ldots, r_G\}$, it calculates a group-relative advantage $\hat{A}_i = \frac{r_i - \text{mean}(\mathbf{r})}{\text{std}(\mathbf{r})}$ for each output $o_i$. By combining a KL-divergence penalty against a reference model $\pi_{\text{ref}}$, it constructs a PPO-clipped objective function $\mathcal{J}_{\text{GRPO}}(\theta)$, which is then maximized to update the LLM parameters $\theta$.

Training-Free GRPO repurposes the core logic of such group relative policy optimization, but translates it into a context-based gradient-free process. Instead of instantiating the RL *policy* as a tunable LLM $\pi_\theta$, our *policy* $\pi_{\theta, \mathcal{E}}$ is a permanently frozen LLM with a variable *experiential knowledge* $\mathcal{E}$ initialized to $\emptyset$ in the context.

**Rollout and Reward.** As shown in Figure 2, our rollout and reward process mirrors that of GRPO exactly. Given a query $q$, we perform a parallel rollout to generate a group of $G$ trajectories or outputs $\{o_1, o_2, \ldots, o_G\}$ by directly injecting all the current experiences $\mathcal{E}$ into the context, i.e., $\pi_{\theta, \mathcal{E}}(o_i|q)$. Identical to the above standard GRPO setup, we score each output $o_i$ by the reward model $\mathcal{R}$ to obtain a scalar reward $r_i = \mathcal{R}(o_i, y)$.

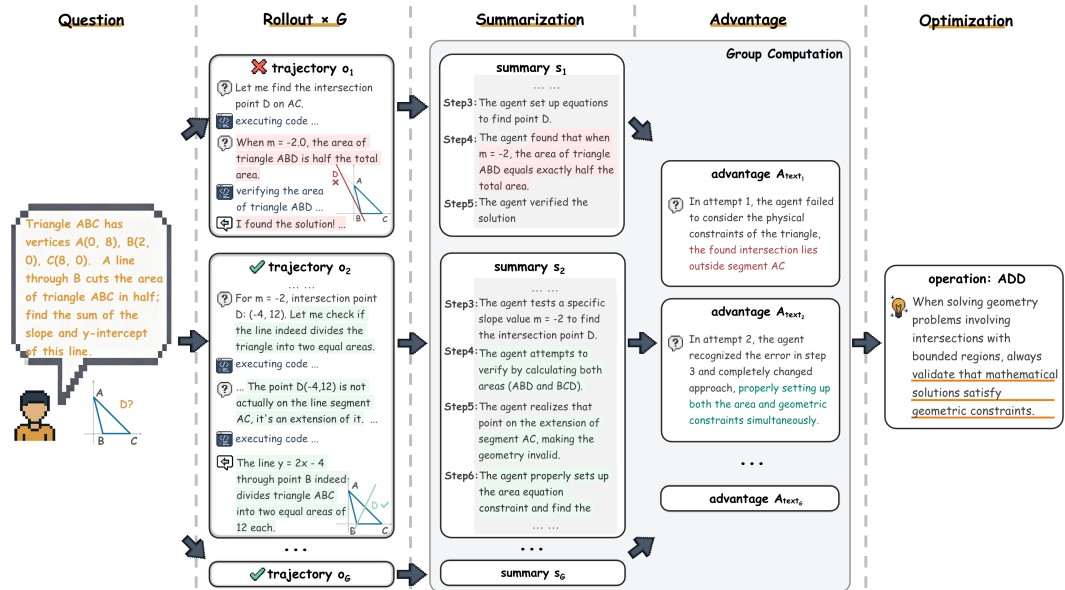

Figure 3: Example of a Training-Free GRPO learning step.

**Group Advantage Computation.** To provide an optimization direction for policy LLM parameters $\theta$, vanilla GRPO computes a numerical advantage $\hat{A}_i$ that quantifies each output $o_i$'s relative quality within its group. Specially, when all $G$ outputs from a group receive identical rewards (i.e., $\text{std}(\mathbf{r}) = 0$), $\hat{A}_i = 0$ and no optimization direction will be provided. Similarly, Training-Free GRPO performs an analogous comparison between outputs within each group, but produces a *semantic group advantage* $A_{\text{text}_i}$ in the form of natural language, as shown in Figure 3. $A_{\text{text}_i}$ articulates the reasons for the relative success or failure of output $o_i$, functionally equivalent to vanilla GRPO's $\hat{A}_i$, delivering the optimization direction of what actions could lead to high rewards. Also, for groups with $\text{std}(\mathbf{r}) = 0$, $A_{\text{text}_i}$ will not be generated due to the lack of optimization direction. Specifically, for each output $o_i$, we first ask the same frozen LLM $\mathcal{M}_\theta$ to provide a step-by-step summary $s_i = \mathcal{M}_\theta(p_{\text{summary}}, q, o_i, y)$, where $p_{\text{summary}}$ is a prompt template that incorporates the query $q$, output $o_i$ and ground truth $y$. With such summaries $\mathbf{s} = \{s_1, s_2, \ldots, s_G\}$, the LLM $\mathcal{M}_\theta$ then extracts the $A_{\text{text}_i} = \mathcal{M}_\theta(p_{\text{adv}}, q, i, \mathbf{s}, y, \mathbf{r})$ for each output $o_i$, where $p_{\text{adv}}$ is the prompt template for advantage generation.

**Optimization.** Whereas vanilla GRPO optimizes the LLM policy $\pi_\theta$ via gradient ascent on $\mathcal{J}_{\text{GRPO}}(\theta)$ computed by all advantages $\hat{A}_i$ in a single batch, we freeze the LLM parameter $\theta$ in our policy $\pi_{\theta, \mathcal{E}}$ and optimize the *experiential knowledge* $\mathcal{E}$ using all semantic advantages $A_{\text{text}_i}$ from the current batch. Specifically, given existing $\mathcal{E}$, the same frozen LLM $\mathcal{M}_\theta$ generates a list of operations, where each operation could be:

- *Add:* Directly append a new experience inspired by $A_{\text{text}_i}$ to the experiential knowledge $\mathcal{E}$.

- *Delete:* Remove a low-quality experience from $\mathcal{E}$ according to $A_{\text{text}_i}$.

- *Modify:* Refine or improve an existing experience in $\mathcal{E}$ based on insights from $A_{\text{text}_i}$.

- *Keep:* The *experiential knowledge* $\mathcal{E}$ remains unchanged.

Similar to vanilla GRPO, we run the above process for multiple epochs, where each epoch may contain several optimization batches. In each batch, after updating the *experiential knowledge* $\mathcal{E}$, the policy $\pi_{\theta, \mathcal{E}}(\cdot|q)$ produces a shifted output distribution in subsequent learning batches. This mirrors the effect of the GRPO policy LLM update by steering the parameters $\theta$ towards higher-reward outputs, but achieves this by altering the *experiential knowledge* $\mathcal{E}$ in the context rather than the LLM parameters. And our frozen LLM parameters $\theta$ acts as a strong prior, ensuring output coherence and providing a built-in stability analogous to the KL-divergence constraint in GRPO that prevents the policy from deviating excessively from $\pi_{\text{ref}}$.

Table 1: Learning cost and evaluation performance of agentic tool-use Reinforcement Learning (RL) methods on AIME benchmarks (Mean@32, %) and WebWalker QA (Average accuracy, %).

| LLM | Method | Training Set | Cost | AIME'24 | AIME'25 | WebWalker |
|-----|--------|--------------|------|---------|---------|-----------|
| DeepSeek-V3.1-Terminus | ReAct | - | - | 80.0 | 67.9 | 67.5 |
| | ReAct+Ours | DAPO-100 | ≈$18 | 82.7 (↑2.7) | 73.3 (↑5.4) | 58.8 |
| | | AFM-100 | ≈$40 | 79.6 | 68.1 | 71.0 (↑3.5) |
| Qwen2.5-32B-Instruct | ReAct | - | - | 31.8 | 25.5 | 26.6 |
| | ReAct+Ours | DAPO-100 | ≈$0.3 | 34.2 (↑2.4) | 28.4 (↑2.9) | 30.3 |
| | | AFM-100 | ≈$1.8 | 29.6 | 24.5 | 32.4(↑5.8) |
| | Retool | DAPO-100 | ≈$1,000 | 43.2 | 35.3 | 31.9 |
| | MiroThinker | AFM-100 | ≈$1,200 | 20.8 | 13.9 | 35.8 |

Table 2: Learning cost and Mean@32 (%) of tool-free RL methods on AIME benchmarks.

| LLM | Method | Training Set | Cost | AIME'24 | AIME'25 |
|-----|--------|--------------|------|---------|---------|
| DeepSeek-V3.1-Terminus | Direct Prompting | - | - | 68.6 | 52.9 |
| | Training-Free GRPO | DAPO-100 | ≈$8 | 72.6 (↑4.0) | 54.0(↑1.1) |
| | GRPO Training | DAPO-100 | ≈$5,000 | 75.7 | 57.1 |
| Qwen2.5-32B-Instruct | Direct Prompting | - | - | 16.4 | 13.2 |
| | Training-Free GRPO | DAPO-100 | ≈$0.2 | 16.8 (↑0.4) | 13.8 (↑0.6) |
| | GRPO Training | DAPO-100 | ≈$800 | 20.3 | 14.4 |

## 3 EVALUTION

Training-Free GRPO instantiates the Reinforcement Learning (RL) policy as a frozen LLM with variable experiential context for offline learning, and such learned experiences work for In-Context Learning (ICL) during online inference. In this section, we compare Training-Free GRPO against parameter-tuning RL methods in two distinct settings:

- *Agentic Tool-Use Setting* involving math tasks with Python and web search tasks with Google.

- *Tool-Free Setting* assessed on mathematical reasoning tasks.

### 3.1 EXPERIMENTAL SETUP

**Benchmarks.** For mathematical reasoning, we conduct our evaluation on the challenging AIME'24 and AIME'25 benchmarks (AIME, 2025). To ensure robust and statistically reliable results, we evaluate each question with 32 independent runs and report the average Pass@1 score, which we denote as Mean@32. For web searching, we evaluate on the WebWalker QA benchmark (Wu et al., 2025), reporting the average accuracy.

**Methods.** We compare Training-Free GRPO against RL methods that perform gradient-based policy optimization on DeepSeek-V3.1-Terminus (DeepSeek-AI, 2024) and Qwen2.5-32B-Instruct (Yang et al., 2025a) models. For agentic tool-use tasks, we compare against Retool (Feng et al., 2025a) and MiroThinker (Team, 2025), which represent the state of the art in math reasoning and web search, respectively. For tool-free math tasks, we include vanilla GRPO (Shao et al., 2024) as our baseline. All baselines are run with their default hyperparameters and trained to convergence. We run Training-Free GRPO for 3 epochs with batch size of 50, using a group size of 5 for math tasks and 3 for web tasks. The temperature of the frozen LLMs is set to 0.7 during learning and 0.3 during evaluation. To simulate real-world scenarios with limited annotated data, all methods are constrained to 100 training samples. For math, we use a random subset of 100 questions from DAPO-Math-17k (Yu et al., 2025), denoted as DAPO-100. For web search, we use 100 randomly sampled questions from the AFM web interaction RL dataset (Li et al., 2025), denoted as AFM-100.

## 3.2 MAIN RESULTS

**Effectiveness of Training-Free GRPO.** Table 1 and Table 2 summarize the performance of tool-augmented agentic RL and tool-free RL methods, respectively. Whether utilizing the flagship DeepSeek-V3.1-Terminus or the smaller Qwen2.5-32B-Instruct, Training-Free GRPO consistently achieves performance gains over naive baselines, i.e., ReAct (Yao et al., 2023b) in agentic RL settings and direct prompting in tool-free RL scenarios. As shown in Table 1, applying Training-Free GRPO to the frozen DeepSeek-V3.1-Terminus reaches 82.7% on AIME'24, 73.3% on AIME'25, and 71.0% on WebWalker. This represents substantial absolute gains of $+2.7\%$, $+5.4\%$, and $+3.5\%$, respectively, achieved by injecting experiences learned with only 100 out-of-domain samples and zero gradient updates. Crucially, since we only modify in-context prompts for standard inference protocols like direct prompting or ReAct, Training-Free GRPO is distinct from Test-Time Scaling (TTS) methods that introduce new generative mechanisms during inference. Consequently, our approach is orthogonal to TTS and can be seamlessly combined with any TTS strategy during inference. Notably, applying Training-Free GRPO to Qwen2.5-32B-Instruct yields more marginal improvements compared to the flagship DeepSeek-V3.1-Terminus in Table 2. This suggests that the effectiveness of context-based RL optimization is dependent on the underlying model's intrinsic reasoning and introspection capabilities, indicating that certain model capability is a prerequisite for effectively applying Training-Free GRPO.

**Comparison within Identical LLM.** When restricted to an identical LLM, gradient-based RL methods like ReTool and MiroThinker naturally secure higher in-domain performance, benefiting from the extensive search space available via parameter updates. However, as illustrated in Figure 1(b)-(c), Training-Free GRPO effectively interpolates the Pareto frontier, offering a significantly lower learning cost while still delivering meaningful performance improvements. Furthermore, Table 1 reveals a critical limitation of parameter tuning. For example, MiroThinker trained on the web-based AFM-100 dataset suffers a severe performance collapse on mathematical AIME benchmarks. This phenomenon highlights that parameter-based specialization is susceptible to catastrophic forgetting, narrowing the model's capabilities to the training domain at the expense of generalizability. In real-world applications, this would necessitate the high-complexity deployment of multiple specialized models. In contrast, Training-Free GRPO successfully circumvents this issue by maintaining a single, general-purpose frozen LLM. It allows for flexible domain switching simply by plugging the corresponding learned experiences into the context during inference.

**Practical View: Fine-tuning Small Models vs. Prompting Large Models.** In practice, the strategic decision for small teams or cost-sensitive scenarios often lies between fine-tuning a smaller LLM or directly leveraging a flagship model like DeepSeek-V3.1-Terminus. Applying Training-Free GRPO to DeepSeek-V3.1-Terminus offers a far superior solution, which not only slashes learning costs by orders of magnitude from $\$1,000$ to $\$18$ as shown in Table 1, but also yields significantly higher performance than fine-tuned 32B models (82.7% vs. 43.2% on AIME'24 Mean@32). This establishes an accessible, high-performance pathway for real-world applications without the prohibitive infrastructure costs of full model training.

## 3.3 ABLATION ANALYSIS

We perform ablation studies on tool-augmented Training-Free GRPO in both mathematical reasoning and web search scenarios. The results are presented in Figure 4 and Table 3.

**Setup.** To ensure efficiency for ablation studies, we adopt a default group size of $G = 3$ for both math reasoning and web search scenarios. When evaluating on the WebWalker QA benchmark, we use a stratified random sample of 51 instances from the test set, where the sampling is proportionally stratified by difficulty level to guarantee balanced representation across different levels of complexity. Unless specified otherwise, all other hyperparameters remain the same as in Section 3.1.

**Learning Dynamics.** As illustrated in Figure 4, during the 3-epoch learning process, we observe a steady and significant improvement in Mean@3 on the training set. Concurrently, the Mean@32 performance on both AIME'24 and AIME'25 improves with each step, peaking at 81.9% and 71.2% respectively. This demonstrates that experiences learned from only 100 problems generalize effectively to out-of-domain benchmarks. We also observe that the average number of tool calls consistently decreases on both in-domain and out-domain datasets, suggesting that Training-Free GRPO teaches the agent to find shortcuts and use tools more efficiently.

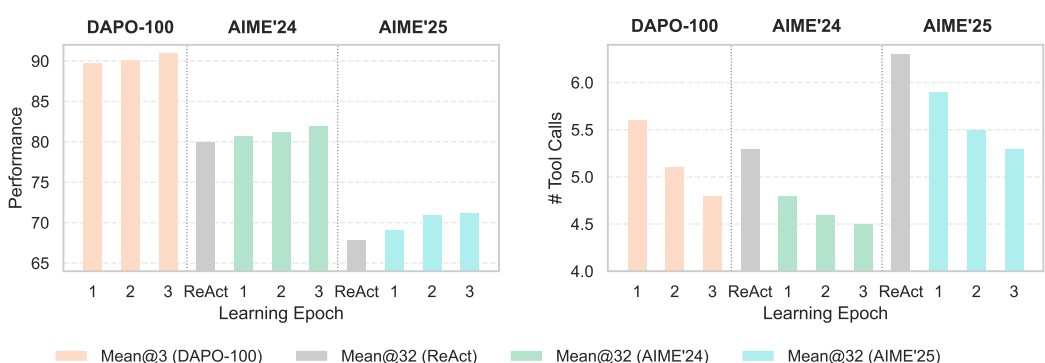

Figure 4: Statistics at each Training-Free GRPO epoch, run on math tasks with tool use and DeepSeek-V3.1-Terminus.

Table 3: Ablation study of Training-Free GRPO on DeepSeek-V3.1-Terminus with tool use, evaluated on AIME benchmarks (Mean@32, %) and WebWalker QA subset (Average accuracy, %).

| Method | AIME24 | AIME25 | WebWalker (subset) |
|---|---|---|---|
| ReAct | 80.0 | 67.9 | 66.7 |
| ReAct + Directly Generated Experiences | 79.8 | 67.3 | 70.6 |
| ReAct + Training-Free GRPO (w/o ground truths) | 80.5 | 68.3 | 72.6 |
| ReAct + Training-Free GRPO (group size $G = 1$) | 80.1 | 68.9 | 72.5 |
| ReAct + Training-Free GRPO (group size $G = 3$) | 81.9 | 71.2 | **74.5** |
| ReAct + Training-Free GRPO (group size $G = 5$) | **82.7** | **73.3** | **74.5** |

**Effectiveness of Learned Experiences.** In Table 3, we compare our method against a baseline where ReAct is enhanced with experiences directly generated by DeepSeek-V3.1-Terminus, matching the format and quantity learned from Training-Free GRPO. Crucially, such directly generated experiences significantly underperform experiences learned by Training-Free GRPO, and even slightly degrade the mathematical ability compared to the ReAct baseline. It highlights that the performance gains of our method stem specifically from the context-based trial-and-error RL process that evolves transferable experiential knowledge.

**Robustness to Reward Signal.** We further evaluate a variant of Training-Free GRPO where ground truth answers $y$ are redacted during the learning process. In this setting, the reward model $\mathcal{R}$ cannot verify correctness of each rollout $o_i$ against $y$, so the semantic group advantage is derived solely by comparing rollouts within each group, forcing the LLM to rely on implicit majority voting and self-consistency. As shown in Table 3, it still improves the ReAct baseline on both math reasoning and web search tasks, demonstrating its robustness and applicability to domains where ground truths are scarce or unavailable.

**Impact of Group Size.** Finally, we analyze the necessity of group-relative computation by setting the group size to $G = 1$, where the LLM is limited to distilling experiences from a single rollout per query, removing the ability to compare diverse trajectories. The results in Table 3 show that $G = 1$ significantly underperforms compared to larger group settings. Moreover, we observe a positive correlation between group size and performance on AIME benchmarks. This confirms that the group-relative mechanism is essential, as larger groups provide a richer context for contrasting successful trajectories against less effective ones, thereby enabling the model to identify and distill more effective experiential knowledge.

## 4 RELATED WORK

This work introduces Training-Free GRPO, which enhances LLM agents by shifting Reinforcement Learning (RL) optimization from the parameter space to the context space. To situate our method, we review the following concepts, analyzing their connections to and distinctions from our approach.

Figure 5: Conceptual comparison of paradigms that improve LLM performance. (a) **Reinforcement Learning (RL)** (Kaelbling et al., 1996): Optimizes a policy which is any state-to-action mapping to maximize reward signals. (b) **In-Context Learning (ICL)** (Brown et al., 2020): Inject examples within the context to help model adapt online without updating model weights. (c) **Test-Time Scaling (TTS)** (Zhang et al., 2025a): Improves performance by allocating more computation during online inference rather than through prior offline optimization.

**LLM Agents.** By leveraging external tools, LLMs can overcome inherent limitations, such as lacking real-time knowledge and precise computation. This has spurred the development of LLM agents that interleave reasoning with actions. Foundational frameworks like ReAct (Yao et al., 2023b) prompt LLMs to generate explicit reasoning and actionable steps, enabling dynamic planning through tool use. Furthermore, Toolformer (Schick et al., 2023) demonstrates that LLMs can learn to self-supervise the invocation of APIs via parameter fine-tuning. Subsequent research has produced sophisticated single- and multi-agent systems, such as MetaGPT (Hong et al., 2024), CodeAct (Wang et al., 2024c), and OWL (Hu et al., 2025), which significantly enhance the quality of planning, action execution, and tool integration. As confirmed in Section 3, our Training-Free GRPO method successfully enhances tool-based LLM agents on both math reasoning and web search tasks, demonstrating its potential for real-world agentic applications.

**Reinforcement Learning.** As shown in Figure 5(a), Reinforcement learning (RL) is a fundamental paradigm that performs trial-and-error (Kaelbling et al., 1996), where an agent interacts, observes and executes actions to maximize reward signals by optimizing a policy. A policy is defined as a mapping from states to actions (Kaelbling et al., 1996), such as simple look-up tables (Gittins et al., 2011) and neural networks (Arulkumaran et al., 2017). Also, RL optimization could be gradient-based strategies or gradient-free methods that search for optimal policies (Arulkumaran et al., 2017). Recent RL studies that enhance LLM performance typically instantiate the policy as a parameterized LLM, and their optimization is usually based on gradient-based updates in the parameter space. For example, Proximal Policy Optimization (PPO) (Schulman et al., 2017) employs a policy model for generation and a separate critic model to estimate token-level value. Group Relative Policy Optimization (GRPO) (Shao et al., 2024) eliminates the need for a critic by estimating advantages directly from groups of responses. Recent research try to apply RL to transform LLMs from passive generators into autonomous agents that learn through environmental interaction. GiGPO (Feng et al., 2025b) implements a two-level grouping mechanism for trajectories, enabling precise credit assignment at both the episode and individual step levels. ReTool (Feng et al., 2025a) uses PPO to train an agent to interleave natural language with code execution for mathematical reasoning. Chain-of-Agents (Li et al., 2025) facilitates multi-agent collaboration within a single model by using dynamic, context-aware activation of specialized tool and role-playing agents. Tongyi Deep Research (Tongyi DeepResearch Team, 2025) introduces synthetic data generation pipeline and conduct customized on-policy agentic RL framework. In this paper, Training-Free GRPO adopts a significantly different way of instantiating the RL policy and the RL optimization process. We instantiate the RL policy as a frozen LLM and variable context, where the experiences within context are iteratively optimized via trial-and-error powered by frozen LLMs without parameter tuning on a separate training set.

**In-Context Learning.** As shown in Figure 5(b), In-Context Learning allows frozen LLMs to learn given only a few examples during online inference (Brown et al., 2020; Dong et al., 2024). Typically, ICL methods organize the examples in the form of input-output demonstration, using various strategies for example selection (Liu et al., 2022), reformatting (Hao et al., 2022) and ordering (Lu et al., 2022). Training-Free GRPO is distinct from such typical few-shot ICL methods, since it does not directly inject input-output examples into the context during online inference. Instead, we include the learned abstract experiential knowledge in the context, which are optimized through trial-and-

error RL process on a separate training set during offline learning phase. Such experiences provide suggestions like "Constraint reduction: When solving constrained approximation problems, first analyze sum/integer constraints to reduce continuous problems to discrete combinatorial selection" for math reasoning. They can guide the frozen LLM for better performance, satisfying the broader idea of ICL, which is to learn from analogy (Dong et al., 2024).

**Test-Time Scaling.** As shown in Figure 5(c), Test-Time Scaling (TTS) is defined as methods that allocate additional computation on test samples during online inference phase (Zhang et al., 2025a), such as Chain-of-Thought (CoT) (Wei et al., 2022) and Tree-of-Thought (ToT) (Yao et al., 2023a). Recent iterative refinement mechanisms during online inference also falls into the concept of TTS (Zhang et al., 2025a), including Self-Refine (Madaan et al., 2023), Reflexion (Shinn et al., 2023), TextGrad (Yuksekgonul et al., 2025), and In-context reinforcement learning (ICRL) (Song et al., 2025; Monea et al., 2024). Self-Refine (Madaan et al., 2023) generates an initial output and then provide verbal feedback for subsequent revisions on the same test sample. Similarly, Reflexion (Shinn et al., 2023) incorporates an external feedback signal for reflection and a new attempt during testing on a single sample. TextGrad (Yuksekgonul et al., 2025) proposes a more general framework, treating optimization as a process of back-propagating textual feedback through a structured computation graph. Recently, In-Context Reinforcement Learning (ICRL) (Song et al., 2025; Monea et al., 2024) demonstrates that LLMs can learn from scalar reward signals by receiving prompts containing their past outputs and associated feedback. A key characteristic of these TTS methods is their focus on iterative, within-sample improvement for a single test sample during online inference. In contrast, Training-Free GRPO optimizes the experiences on a separate training set without accessing any test samples during offline learning, while its online inference remains simple prompting or ReAct, which is orthogonal to TTS and could be combined with any TTS strategies.

**Other Related Methods.** Similar to Training-Free GRPO, several recent studies extract guidelines, templates, or workflows during an offline phase to enhance subsequent training or inference. However, our approach is distinguished by two key factors: *(1) Multi-Round Iterative Optimization*: While prior methods typically extract knowledge in a single pass during offline, Training-Free GRPO treats experiential knowledge as an RL policy, employing multi-epoch learning to iteratively optimize it. *(2) Contrastive Experience Distillation:* Existing methods usually derive insights solely from single successful trajectories, but Training-Free GRPO contrasts multiple successful and failed trajectories for the same query, extracting more robust experiences as validated in Section 3.3. Specifically, ReasonFlux (Yang et al., 2025b) and its variants (Zou et al., 2025; Wang et al., 2025) construct thought templates by analyzing the reasoning behind individual solutions in a single pass. AutoGuide (Fu et al., 2024) generates context-aware guidelines from offline data in a one-pass manner. Agent Workflow Memory (AWM)(Wang et al., 2024d) induces workflows exclusively from successful trajectories and integrates into memory. Finally, Agent KB(Tang et al., 2025) constructs a hierarchical knowledge base using hand-crafted examples and a one-time off-policy learning paradigm, collecting trajectories in the different way of online inference. In contrast, Training-Free GRPO maintains a consistent inference pipeline during offline and online phases, and closely mirrors on-policy RL through multi-epoch iterative updates.

## 5 CONCLUSION

In this paper, we introduced Training-Free GRPO, a novel paradigm that fundamentally rethinks Reinforcement Learning by shifting policy optimization from the parameter space to the context space. By instantiating the policy as a frozen LLM paired with variable experiential knowledge, our method mirrors multi-epoch RL training. We replace the costly gradient updates with the optimization of experiences via semantic group advantages, which are derived from a group of successful and failed trajectories for the same training sample. Empirical evaluations on mathematical reasoning and web search benchmarks demonstrate that Training-Free GRPO establishes a new Pareto frontier between performance and learning cost. Notably, we show that optimizing the context of a flagship DeepSeek-V3.1-Terminus with merely 100 samples outperforms parameter fine-tuned 32B LLMs, while reducing learning costs by orders of magnitude from $800 to $8. By circumventing the risks of catastrophic forgetting and high infrastructure barriers of parameter tuning, Training-Free GRPO establishes a new, highly efficient pathway for adapting powerful LLM agents, making advanced agentic capabilities more accessible and practical for real-world applications.

**Ethics Statement** The present study conforms to the ICLR Code of Ethics. The paper does not involve crowdsourcing nor research with human subjects.

**Reproducibility Statement** All datasets used in the paper are publicly accessible (see Section 3). All the codes are available at `https://anonymous.4open.science/r/Training-Free-GRPO` for reproduction.

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

## A  PROMPTS FOR MATH TASKS

Solve the following problem step by step. You now have the ability to selectively write executable Python code to enhance your reasoning process, e.g., calulating numbers and verifying math computations. Never directly just printing your semantic reasoning in Python. The Python code will be executed by an external sandbox, and the output (returned as a dict with the message in the "message" field) can be returned to aid your reasoning and help you arrive at the final answer. The Python code should be complete scripts, including necessary imports.

Each code snippet is wrapped with
```python
code snippet
```.

The last part of your final response should be in the following format:
<answer> \boxed{The final answer goes here.} </answer>

Figure 6: System prompt for math tasks.

Please solve the problem:
{problem}

When solving problems, you MUST first carefully read and understand the helpful instructions and experiences:
{experiences}

Figure 7: Prompt for supplementing math problems with experiential knowledge $\mathcal{E}$.

An agent system may be provided with some experiences, and then it produces the following trajectory to solve the given problem. Please summarize the trajectory step-by-step:
1. For each step, describe what action is being taken, and which experience has been used in this step.
2. Given the grading of this rollout and the correct answer, identify and explain any steps that represent detours, errors, or backtracking, highlighting why they might have occurred and what their impact was on the trajectory's progress.
3. Maintain all the core outcome of each step, even if it was part of a flawed process.

<trajectory> {trajectory} </trajectory>
<evaluation> {whether the answer is correct or not} </evaluation>
<groundtruth> {the ground truth answer} </groundtruth>

Only return the trajectory summary of each step, e.g.,
1. what happened in the first step and the core outcomes
2. what happened in the second step and the core outcomes
3. ...

Figure 8: Prompt for summarizing each trajectory during Training-free GRPO in math tasks.

An agent system is provided with a set of experiences and has tried to solve the problem multiple times with both successful and wrong solutions. Review these problem-solving attempt and extract generalizable experiences. Follow these steps:

1. Trajectory Analysis:
    − For successful steps: Identify key correct decisions and insights
    − For errors: Pinpoint where and why the reasoning went wrong
    − Note any important patterns or strategies used/missed
    − Review why some trajectories fail? Is there any existing experiences are missed, or experiences do not provide enough guidance?

2. Update Existing Experiences
    − Some trajectories may be correct and others may be wrong, you should ensure there are experiences can help to run correctly
    − You have three options: [modify, add, delete]
        ∗ modify: You can modify current experiences to make it helpful
        ∗ add: You can introduce new experiences to improve future performance
        ∗ delete: You can delete existing experiences
    − You can update at most {`max number of operations`} clear, generalizable lessons for this case
    − Before updating each experience, you need to:
        ∗ Specify when it would be most relevant
        ∗ List key problem features that make this experience applicable
        ∗ Identify similar problem patterns where this advice applies

3. Requirements for each experience that is modified or added.
    − Begin with general background with several words in the experience
    − Focus on strategic thinking patterns, not specific calculations
    − Emphasize decision points that could apply to similar problems

Please provide reasoning in details under the guidance of the above 3 steps. After the step-by-step reasoning, you will finish by returning in this JSON format as follows:

```json
[
    {
        "option": "modify",
        "experience": "the modified experience",
        "modified_from": "G17" # specify the ID of experience that is modified
    },
    {
        "option": "add",
        "experience": "the added experience",
    },
    {
        "option": "delete",
        "delete_id": "the deleted experience ID",
    }, ...
]
```

Note that your updated experiences may not need to cover all the options. You can only use one type of updates or choose to remain all experiences unchanged.

<problem> {`problem`} </problem>
<trajectories> {$G$ `trajectories in the same group`} </trajectories>
<groundtruth> {`answer`} </groundtruth>
<experience> {`experiences`} </experience>

Figure 9: Prompt for semantic group advantage computation based on group rollouts during Training-free GRPO in math tasks.

An agent system is provided with a set of experiences and has tried to solve the problem multiple times. From the reflections, some suggestions on the existing experiences have been posed. Your task is to collect and think for the final experience revision plan. Each final experience must satisfy the following requirements
1. It must be clear, generalizable lessons for this case, with no more than 32 words
2. Begin with general background with several words in the experience
3. Focus on strategic thinking patterns, not specific calculations
4. Emphasize decision points that could apply to similar problems
5. Avoid repeating saying similar experience in multiple different experiences

<experience> {experiences} </experience>
<suggested_updates> {group advantage} </suggested_updates>

Please provide reasoning in each of the suggestions, and think for how to update existing experiences
You have two update options: [modify, merge]
- modify: You can modify current experiences to make it helpful - merge: You can merge some similar experiences into a more general forms to reduce duplication
After generating the step-by-step reasoning, you need to give the final experience revision details by returning in this JSON format as follows:
```json
[
    {
        "option": "modify",
        "experience": "the modified experience",
        "modified_from": "G17" # specify the ID of experience that is modified
    },
    {
        "option": "merge",
        "experience": "the merged experience",
        "merged_from": ["C1", "C3", "S4", ...] # specify the str IDs of experiences that is merged from,
at least 2 IDs are needed
    },
    {
        "option": "delete",
        "delete_id": "the deleted experience ID",
    }, ...
]
```

Note that your updated experiences may not need to cover all the options. You can only use one type of updates or choose to remain all experiences unchanged.

<problem> {problem} </problem>
<trajectories> {$G$ trajectories in the same group} </trajectories>
<groundtruth> {answer} </groundtruth>

Figure 10: Prompt for optimizating experiential knowledge $\mathcal{E}$ based on semantic group advantages in the same batch during Training-free GRPO on math tasks.

# B CASE STUDY

## B.1 EXPERIENCE-GUIDED TOOL-INTEGRATED MATH REASONING

We consider a geometric configuration with two rectangles $ABCD$ and $EFGH$ where $D, E, C, F$ are collinear in that order, and $A, D, H, G$ are concyclic. Given $BC = 16$, $AB = 107$, $FG = 17$, $EF = 184$, the task is to determine $CE$.

**ReAct Baseline without Experiences.** As shown in Figure 11, the unassisted agent initializes a coordinate system with $D = (0,0)$ and models $ABCD$ as axis-aligned. For $EFGH$, it places $E = (x_f - 184, 0)$, $F = (x_f, 0)$ and, critically, sets $G = (x_f, +17)$, $H = (x_f - 184, +17)$, i.e., with a positive vertical orientation for the short side. It then enforces the four-point concyclicity of $A(0, 16)$, $D(0, 0)$, $G$, $H$ via a determinant condition and solves for $x_f$, yielding $x_f \approx 184.092$ and consequently $x_e = x_f - 184 \approx 0.092$. From this, it reports $CE \approx 106.9076$ and an "exact" expression $199 - \sqrt{8481}$.

This trajectory exhibits three systemic issues: (i) misinterpretation of the vertical orientation (wrong sign for the $y$-coordinates of $G, H$), (ii) inconsistent handling of the order $D$–$E$–$C$–$F$ and the lack of a unified parameterization for segment relations, and (iii) absence of systematic, comprehensive post-solution verification—i.e., no integrated check that the final coordinates simultaneously satisfy rectangle dimensions. These issues lead to an incorrect cyclic constraint (e.g., an intermediate relation of the form $x(x + 184) = 17$) and acceptance of a spurious solution without full geometric verification. Note that although $CE \approx 106.91$ lies within $0 < CE < 107$, this alone does not validate the solution; the critical failure was the lack of holistic consistency checks across all problem constraints.

---

**Tool-integrated ReAct on Math Reasoning without Experiences**

**Question**
Rectangles $ABCD$ and $EFGH$ are drawn such that $D, E, C, F$ are collinear. Also, $A, D, H, G$ all lie on a circle. If $BC = 16, AB = 107, FG = 17$, and $EF = 184$, what is the length of $CE$?

- - - - - - - - - - - - - - - - - - - - - - - - - - - - - - - - - - - - - - - - - - - - - - - - - - - -

**Trajectory**
[Agent] I'll place point $D$ at the origin $(0, 0)$ and use a horizontal line for $D, E, C, F$. For rectangle $ABCD$, set $A = (0, 16)$, $B = (107, 16)$, $C = (107, 0)$, $D = (0, 0)$.
[Tool] Set coordinates accordingly.
[Agent] Because $EFGH$ is a rectangle with $EF$ horizontal and $FG = 17$, set $F = (x_f, 0)$, $E = (x_f - 184, 0)$, and initially set $G = (x_f, +17)$, $H = (x_f - 184, +17)$.
[Agent] Impose concyclicity of $A(0, 16), D(0, 0), G(x_f, 17), H(x_f - 184, 17)$ using the 4-point determinant condition, derive an equation in $x_f$.
[Tool] Solve the resulting quadratic for $x_f$; obtain $x_f \approx 184.092$ and thus $x_e = x_f - 184 \approx 0.092$.
[Agent] Compute $CE = |107 - x_e| \approx 106.9076$.
[Agent] Swap point orderings ($D$–$C$–$E$–$F$, $E$–$D$–$C$–$F$) and re-derive. At one point, use an angle-sine relation and get $x_e(x_e + 184) = 17$ leading to $x_e = -92 + \sqrt{8481} \approx 0.0923$, hence $CE \approx 106.9076$.
[Agent] Present an "exact form" $CE = 199 - \sqrt{8481}$ as the final answer.

**Outcome**
Incorrect: $CE \approx 106.9076$ (reported as $199 - \sqrt{8481}$).

---

Figure 11: Example trajectory without experience of ReAct on math reasoning tasks.

---

**Tool-integrated ReAct on Math Reasoning with Experiences**

**Question**
Rectangles $ABCD$ and $EFGH$ are drawn such that $D, E, C, F$ are collinear. Also, $A, D, H, G$ all lie on a circle. If $BC = 16, AB = 107, FG = 17$, and $EF = 184$, what is the length of $CE$?

**Experiences**
[1] When solving geometry problems with intersections, validate solutions lie within bounded regions or segments, not on extensions, to avoid extraneous answers.
...
[7] When solving geometry with constraints (e.g., trapezoids), validate all conditions: parallel sides via slopes, convexity via cross products, to avoid extraneous solutions.
...
[29] When constructing points on line extensions, carefully interpret directional phrases like 'beyond B'.
...
[37] In geometry problems with points on sides of a triangle and given segment lengths, first determine all three side lengths by summing the appropriate segments. For example, if points on AB, AC, BC are given with AM, MB, AN, NC, BP, PC, then AB = AM + MB, AC = AN + NC, BC = BP + PC. This ensures the triangle is consistently defined before coordinate placement or area calculations.
...

- - - - - - - - - - - - - - - - - - - - - - - - - - - - - - - - - - - - - - - - - - - - -

**Trajectory**
[Agent] Fix the order $D$–$E$–$C$–$F$ on a horizontal line. Set $CE = x$ with the segment constraint $0 < x < 107$. (By [G29],[G1])
[Tool] Record relations: $DE = 107 - x$ and $CF = 184 - x$ (from [G37]).
[Agent] Place $D = (0,0)$, $E = (107 - x, 0)$, $C = (107, 0)$, $F = (291 - x, 0)$, ensuring $EF = 184$.
[Agent] Choose the vertical orientation consistently: since $A = (0, 16)$ and $D = (0, 0)$, set the short side $FG = 17$ downward, i.e., $H = (107 - x, -17)$ and $G = (291 - x, -17)$. (This corrects the sign error in the no-experience path)
[Agent] Impose concyclicity using the circle equation $x^2 + y^2 + Dx + Ey + F = 0$. With $A(0, 16)$ and $D(0, 0)$, get $F = 0$, $E = -16$.
[Tool] Substitute $H(107 - x, -17)$ and $G(291 - x, -17)$ to obtain two equations. Subtract to eliminate constants and get $D = 2x - 398$.
[Agent] Substitute back to get the quadratic $x^2 - 398x + 30576 = 0$.
[Tool] Compute the discriminant: $398^2 - 4 \cdot 30576 = 36100 = 190^2$. Roots: $x = 294$ and $x = 104$.
[Agent] Apply the segment constraint $0 < x < 107$ to discard $x = 294$ and select $x = 104$ (by [G1])
[Agent] (Full verification by [G7]) $DE = 107 - 104 = 3$, $CF = 184 - 104 = 80$, $EF = 184$, $FG = 17$. The circle $x^2 + y^2 - 190x - 16y = 0$ passes through $A = (0, 16), D = (0, 0), H = (3, -17), G = (187, -17)$. All constraints satisfied.

**Outcome**
Correct: $CE = 104$.

---

Figure 12: Example trajectory with experiences in Math Reasoning

**ReAct Enhanced with Experiences Learned from Training-Free GRPO.** Refer to Figure 12, with a curated experience pool, the agent follows a structured pipeline:

- Directional ordering ([29]) and boundedness validation ([1]): It fixes the order $D$–$E$–$C$–$F$ on a line and sets $CE = x$ with $0 < x < 107$, ensuring $E$ lies on segment $DC$ and $F$ lies beyond $C$.

- Segment-addition parameterization ([37]): It uses $DE + EC = DC = AB = 107$ and $EC + CF = EF = 184$ to obtain $DE = 107 - x$, $CF = 184 - x$, and places $D = (0,0)$, $E = (107 - x, 0)$, $C = (107, 0)$, $F = (291 - x, 0)$.

- Consistent vertical orientation and cyclic modeling: Noting $A = (0, 16)$, $D = (0, 0)$, it orients the short side downward ($FG = 17$) so $H = (107 - x, -17)$, $G = (291 - x, -17)$. Using the circle equation $x^2 + y^2 + Dx + Ey + F = 0$ with $A$ and $D$ yields $F = 0$, $E = -16$. Substituting $H$ and $G$, subtracting the two equations gives $D = 2x - 398$; back-substitution reduces to the quadratic $x^2 - 398x + 30576 = 0$, with discriminant $398^2 - 4 \cdot 30576 = 36100 = 190^2$ and roots $x = 104, 294$.

- Root selection and full verification ([1], [7]): Applying $0 < x < 107$ filters out $x = 294$, selecting $x = 104$. The agent then verifies all constraints: $DE = 107 - 104 = 3$, $CF = 184 - 104 = 80$, $EF = 184$, $FG = 17$, and confirms that the circle $x^2 + y^2 - 190x - 16y = 0$ passes through $A = (0, 16)$, $D = (0, 0)$, $H = (3, -17)$, $G = (187, -17)$.

**Comparative Analysis.** This case reveals a clear causal link between experience-guided behaviors and correctness. Experience [29] eliminates directional ambiguity and enforces the correct collinearity order, directly addressing the baseline's misplacement of $G$, $H$. Experience [37] induces a clean single-variable parameterization ($DE = 107 - x$, $CF = 184 - x$), which simplifies the cyclic constraint to a solvable quadratic. Experience [1] imposes a necessary boundedness filter ($0 < x < 107$) to discard extraneous roots. Finally, experience [7] mandates comprehensive post-solution verification (rectangle dimensions, collinearity, concyclicity), preventing acceptance of spurious solutions. Compared to the unassisted trajectory, the experience-informed reasoning corrects the vertical orientation, resolves ordering and parameterization inconsistencies, and installs principled validation gates. This case demonstrates the positive impact of integrating domain-specific experiences on reliability and accuracy in tool-integrated mathematical reasoning.

## B.2 EXPERIENCE-GUIDED WEB SEARCHING

We consider a web searching task from WebWalkerQA: quantify 2024 rewards for (i) creators in the Creator Program (weekly amount), and (ii) players in the Play-2-Airdrop during Private Beta 4 (total pool and duration).

**ReAct Baseline without Experiences.** As summarized in Figure 13, the unassisted agent issues multiple broad searches and relies heavily on result snippets and third-party summaries, delaying clicks into authoritative pages. It eventually opens the PB4 registration post to confirm "Play-2-Airdrop Campaign: 200,000 ILV over six months," but continues to scan snippets for the Creator Program value without opening the relevant official post. Connection errors to one official page cause the agent to revert to broad searches rather than alternative primary-source strategies (e.g., site-specific queries or adjacent official posts). The trajectory remains incomplete: it reports the Play-2-Airdrop figure but fails to confirm the Creator Program's "100 ILV weekly" from an official source, yielding an incorrect/incomplete answer.

**ReAct Enhanced with Experiences Learned from Training-Free GRPO.** Refer to Figure 14. With a curated experience pool, the agent follows a disciplined pipeline: (1) prioritize official sources ([14], [4]) and open the PB4 overview and registration posts to extract the "200,000 ILV over six months" and Testnet/Mainnet allocation structure; (2) refine search terms to target formal titles ([2], [12], [1]) and open "The Illuvium Creator Program Is Getting An Upgrade!"; (3) extract the exact line "The prize pool of 100 ILV tokens awaits participating Creators each week," and (4) synthesize both verified statements into a complete answer aligned with the question requirements ([25]). This results in the correct, fully supported output: creators earn 100 ILV weekly; players have a 200,000 ILV pool distributed over six months in PB4's Play-2-Airdrop.

**Comparative Analysis.** Experience-guided behaviors directly address baseline deficiencies: primary source prioritization ([14], [4]) removes reliance on snippets and third-party coverage; document identification ([1]) and iterative refinement ([2], [12]) ensure the agent locates and opens the

exact Creator Program post; partial match evaluation ([25]) steers the agent to confirm numerical claims at their authoritative origin. In contrast, the baseline wastes context on searches without content acquisition, leaves critical values unverified, and produces an incomplete answer.

---

**Tool-integrated ReAct on Web Search without Experiences**

**Question**
In 2024, how much ILV reward can creators participating in the Creator Program and players engaging in the Play-2-Airdrop during Private Beta 4 on the Illuvium platform earn?

- - - - - - - - - - - - - - - - - - - - - - - - - - - - - - - - - - - - - - - - - -

**Trajectory**
[Agent] I'll start with a broad web search for both program names and rewards in 2024.
[Tool] google_search("Illuvium Creator Program Play-2-Airdrop Private Beta 4 2024 rewards ILV")
[Agent] From snippets, PB4 mentions a "Play-2-Airdrop campaign" and "200,000 ILV over six months." This likely answers half the question. (continue searching rather than opening)
[Agent] I'll search specifically for the Creator Program rewards.
[Tool] google_search("Illuvium Creator Program 2024 rewards ILV allocation")
[Agent] I see references (Binance Square, media blogs) to "100 ILV per week," but can't confirm from illuvium.io quickly. I'll keep searching.
[Agent] I'll fetch the PB4 registration page to confirm the Play-2-Airdrop details.
[Tool] get_content("https://illuvium.io/news/register-now-for-private-beta-4")
[Agent] Extracted: "Play-2-Airdrop Campaign: 200,000 ILV tokens over six months." That confirms the second quantity.
[Agent] I'll look for the Airdrop Season details page.
[Tool] google_search("Illuvium Airdrop Season Details blog post 2024")
[Tool] get_content("https://illuvium.io/news/illuvium-airdrop-season-details")
[Agent] More searches for "Illuvium 100 ILV per week creator rewards April 2024" and "site:illuvium.io Creator Program 100 ILV" yield sparse official confirmation; I'll defer the creator number.
[Agent] Synthesis: I will answer with what's confirmed officially (Play-2-Airdrop 200,000 ILV over 6 months) and note the creator reward as unclear.

**Outcome**
Creators: (not clearly specified; mentions unverified "100 ILV per week"). Players (PB4 Play-2-Airdrop): 200,000 ILV over six months.

---

Figure 13: Example trajectory without experiences in Web Searching

---

**Tool-integrated ReAct on Web Search with Experiences**

**Question**

In 2024, how much ILV reward can creators participating in the Creator Program and players engaging in the Play-2-Airdrop during Private Beta 4 on the Illuvium platform earn?

**Experiences**

[1] Document identification: Focus on extracting formal titles and collection names from official metadata and headers rather than inferring relationships from content descriptions.

[2] Systematic exploration: Progress from broad initial queries to targeted content extraction using adaptive search strategies that balance discovery with precision.

...

[4] Source prioritization: Prioritize systematic extraction from authoritative comprehensive documents over fragmented information for coherent topic coverage.

...

[12] Iterative refinement: Continuously refine search terms based on emerging patterns while periodically re-evaluating previously encountered information to avoid overlooking correct solutions and maintain essential contextual elements.

...

[14] Primary source prioritization: Prioritize locating official primary source documents over secondary coverage to ensure accuracy and capture exact language for definitive metadata and hierarchical relationships.

...

[25] Partial match evaluation: Systematically evaluate partial matches from initial searches against all criteria before pursuing alternative paths.

...

- - - - - - - - - - - - - - - - - - - - - - - - - - - - - - - - - - - - - - - -

**Trajectory**

[Agent] Goal breakdown: Two official numbers required: (1) Creator Program weekly ILV; (2) PB4 Play-2-Airdrop total ILV and time frame.

[Agent] Start with broad search, then pivot to official illuvium.io pages for PB4 and Creator Program. (By [2], [14], [4])

[Tool] google_search("Illuvium Creator Program rewards 2024 Private Beta 4 ILV earnings", num_results=5)

[Agent] Identify official pages in results: "Illuvium Beta 4: All You Need to Know" and "Register Now for Private Beta 4". Open them to confirm Play-2-Airdrop details. (By [14], [25])

[Tool] get_content("https://illuvium.io/news/illuvium-beta-4-all-you-need-to-know")

[Agent] Extracted from official PB4 overview: Testnet phases get 10% of the Play-2-Airdrop pool; clarifies structure and timeline.

[Tool] get_content("https://illuvium.io/news/register-now-for-private-beta-4")

[Agent] Extracted explicit official number: "Play-2-Airdrop Campaign: 200,000 ILV tokens" over six months. This covers the player reward for PB4 context.

[Agent] Iteratively refine search for the Creator Program exact weekly reward using formal title cues like "Creator Program ... upgrade". (key behavior; By [12], [1])

[Tool] google_search("Illuvium Creator Program rewards ILV 2024", num_results=5)

[Agent] Click the official post: "The Illuvium Creator Program Is Getting An Upgrade!". (By [14], [1], [4])

[Tool] get_content("https://illuvium.io/news/illuvium-creator-program-is-getting-an-upgrade")

[Agent] Extracted exact statement from official source: "The prize pool of 100 ILV tokens awaits participating Creators each week." This provides the creator reward.

**Outcome**

Creators can earn 100 ILV weekly in the Creator Reward Program, and 200,000 ILV will be distributed over six months for players in the Play-2-Airdrop during Private Beta 4.

Figure 14: Example trajectory with experiences in Web Searching

Table 4: Mean@32 (%) of few-shot In-Context Learning (ICL) on AIME benchmarks using DeepSeek-V3.1-Terminus.

| Method | Context | AIME'24 | AIME'25 |
|--------|---------|---------|---------|
| Direct Prompting | - | 68.6 | 52.9 |
| Training-Free GRPO | Experiences learned on DAPO-100 | 72.6 (↑4.0) | 54.0 (↑1.1) |
| Few-shot ICL | Retrieved from DAPO-Math-17k | 67.6 | 45.4 |

## C  COMPARISON WITH FEW-SHOT IN-CONTEXT LEARNING METHOD

In this appendix, we compare Training-Free GRPO against few-shot ICL baseline to evaluate the experiential context derived from trial-and-error.

**Benchmarks.** We conduct our evaluation on the challenging AIME'24 and AIME'25 benchmarks (AIME, 2025). To ensure robust and statistically reliable results, we evaluate each question with 32 independent runs and report the average Pass@1 score, which we denote as Mean@32.

**Setup.** We conduct ICL experiments using DeepSeek-V3.1-Terminus (DeepSeek-AI, 2024) as a frozen, text-only LLM without tool use. Our evaluation compares: (1) zero-shot direct prompting, and (2) a few-shot ICL baseline that uses the Qwen3-Embedding-8B model (Zhang et al., 2025b) to retrieve top-3 similar questions from the DAPO-Math-17K dataset (Yu et al., 2025). Each retrieved example includes both the question and a verified step-by-step solution trajectory generated by DeepSeek-V3.1-Terminus. For Training-Free GRPO, we use a group size of 5 and randomly sample only 100 questions from DAPO-Math-17k denoted as DAPO-100, learning with 3 epochs and a batch size of 50. The final optimized experiences are then used in context for the AIME test questions.

**Results.** As presented in Table 4, the experiences distilled by Training-Free GRPO exhibit clear superiority over both direct prompting and the standard few-shot ICL baseline. This demonstrates that our approach effectively steers model behavior by injecting transferable problem-solving heuristics into the context. Remarkably, although these experiences are derived from trial-and-error on merely 100 out-of-domain training samples, they prove more potent for guiding complex reasoning than step-by-step demonstrations retrieved from the extensive DAPO-Math-17k dataset.

## D  THE USE OF LARGE LANGUAGE MODELS

We clarify that no LLMs were employed in the writing or polishing of this paper. All content presented herein is the result of original research and critical evaluation by the authors.

