# OpenReview forum: "Training-Free Group Relative Policy Optimization"
_ICLR.cc/2026/Conference — Submitted to ICLR 2026_

### Official Review · Reviewer_zSXk · 2025-10-24

**Soundness:** 2
**Presentation:** 2
**Contribution:** 2
**Rating:** 2
**Confidence:** 4

**Summary:**

This work proposes Training-free GRPO, a method that keeps model weights fixed while introducing an external experience library to update experiences. In each round, groups of rollouts per query are sampled, and the LLM summarizes winners and losers to extract a natural-language “semantic” advantage. This information is then used to update an external experience library, which is later fed back as a token prior in subsequent API calls. Essentially, the authors introduce a way to build an internal RAG system via prompt engineering and demonstrate that such an internal RAG can improve model performance on AIME and WebWalkerQA.

**Strengths:**

- The idea is clear and could work under specific model setups.

- The overall paper is well written, with clear motivation and rationale.

- The method is practical and can be replicated by the community.

**Weaknesses:**

- Please reconsider how you classify your method. Labeling it as "RL" is misleading, given that the model parameters remain frozen and the reward mechanism is not clearly defined. A more appropriate framing would be under prompt-based RAG or inference-time memory optimization. Additionally, since there is no performance comparison with RL baselines, labeling your method as "RL" in all tables is inappropriate. Finally, emphasizing only the training cost savings without comparing performance against GRPO is insufficient and could weaken the overall claim.

- The experience may accumulate over time, eventually exceeding the model's context window, which suggests that this method does not scale well. Moreover, the proposed framework appears unsuitable for training smaller models with limited context and rollout lengths; for example, Qwen-Math with only a 4K context window.

- Only a limited set of baselines is compared, as ReAct is the only method evaluated under the same base model and training configuration. Moreover, the results presentation and comparisons are **improper and unfair** (see Q1 and Q2 for details).

- There is a limitation in the benchmarking setup, as only one benchmark from the math domain (AIME) and one from the web domain are presented. This does not sufficiently support the paper’s claims or demonstrate the effectiveness of the proposed method through empirical evidence. It is recommended to include results on at least two classic math benchmarks, such as **AMC23**, **Minerva**, or **Olympiads**, to strengthen the evaluation. Additionally, regarding the web domain, it would be helpful to clarify whether your method can be applied to more actual, real-world web tasks such as **WebArena** or **WebShop**, as WebWalkerQA is quite limited to reflect the actual web browsing & exploring ability.

- The idea and framework is similar to previous work such as Reflexion and Self-Refine, even though it's discussed.

**Questions:**

[Q1] Comparison in Table 1 is improper, you are not doing your method on Qwen2.5-32B-Instruct, but why you put those results and compare with your method on DeepSeek-V3.1?

[Q2] Comparison in Table 5 is also unfair, please avoid comparing your method towards the other methods on different base models. Make sure they are trained on the same base model with the same training dataset configuration.

[Q3] How would it work on smaller model? Please report training result over model less than 10B.

Please also addresses the concerns in weakness part.

---

> ### Author Response · Authors · 2025-12-03
>
> We appreciate your thoughtful feedback. We would like to clarify the points you raised as follows:
>
> ## For Weakness 1 and 3, Question 1 and 2
>
> As discussed in the revised Section 4, Training-Free GRPO is a RL method during offline learning, and it is also an ICL method during online inference. But it is not a Test-Time Scaling method.
> - Reinforcement Learning (RL): Following the fundamental definition [1], RL is a paradigm of learning through trial-and-error to maximize rewards by optimizing a policy. A policy is defined as a mapping from states to actions [1], which could be simple look-up tables [2]. Also, RL optimization could be gradient-free methods [3].
> In the Section 1 of the revised manuscript, we clarify that Training-Free GRPO satisfies this fundamental definition of RL. It instantiates the RL policy as the union of a frozen LLM and its variable context, as illustrated in the revised Figure 1(a). Also, during its multi-epoch learning process, the policy context is optimized through trial-and-error to maximize rewards.
> - In-Context Learning (ICL): Training-Free GRPO include the learned experiential knowledge in the context during online inference, satisfying the broader idea of ICL, i.e., learn from analogy [4].
> - Distinction from Test-Time Scaling (TTS): TTS allocates additional computation for each test sample (e.g., Tree of Thoughts, Self-Refine) during online inference or testing [5]. In contrast, Training-Free GRPO optimizes experiences on a separate training set without access to any test samples, as shown in the revised Figure 1(a). Once these optimized experiences are incorporated into the context, online inference proceeds with simple prompting or ReAct, requiring no extra computation at test time. In essence, the method differs from standard prompting or ReAct only through its use of learned static context. and does not require any additional computation during testing. Thus, Training-Free GRPO is orthogonal to TTS methods and can be readily combined with any TTS strategy.
>
> In this paper, we focus on practical scenarios of adapting LLMs to specialized tasks. Regardless of model size, the central questions we aim to answer are: Which approach offers the best cost-efffectiveness for real-world deployment? As a strategic decision for our own business, which one promises a better return on investment? As confirmed in Figure 1(b)-(c) of the revised manuscript, comparing to the RL-trained 32B LLM, Training-Free GRPO achieves higher performance with much lower learning costs.
>
> We also agree with your suggestion of comparing with GRPO and more baselines with the same base model. In Section 3.2 of the revised manuscript, we compare Training-Free GRPO against multiple vanilla RL methods on identical LLMs, including both Qwen2.5-32B-Instruct and DeepSeek-V3.1-Terminus. As expected, parameter tuning offers a larger search space and typically yields higher performance ceilings than optimizing prompts. However, parameter-based specialization is susceptible to catastrophic forgetting, narrowing the model’s capabilities to the training domain at the expense of generalizability. In contrast, Training-Free GRPO circumvents this issue by maintaining a single, general-purpose frozen LLM. It allows for flexible domain switching simply by plugging the corresponding learned experiences.
>
> ## For Weakness 2 and Question 3
>
> As pointed out, the experiences may accumulate over time. Our experiments show a performance decline at the $5^{th}$ learning epoch. Though it does not exceed the model's context window, accumulating excessive experiences may eventually saturate the context or introduce noise. And in practice, it is necessary to halt optimization at the appropriate epoch, such as epoch $3$ in our experiments.
>
> As for the applicability of small models, we explicitly point out this in the revised Section 3.2. The effectiveness of context-based RL optimization is dependent on the underlying model’s intrinsic reasoning and introspection capabilities, indicating that certain model capability is a prerequisite for effectively applying Training-Free GRPO.
>
> ## For Weakness 4
>
> Thanks for you suggestion. Due to limited time, our future work will compare Training-Free GRPO against the baselines on these benchmarks.

---

> > ### Author Response · Authors · 2025-12-03
> >
> > ## For Weakness 5
> >
> > We have improved the discussion of related work for clarity, which is now presented in Section 4 of the revised manuscript. According to the recent survey [5], iterative refinement methods like Reflexion and Self-Refine fall into the concept of Test-Time Scaling (TTS), which is defined as methods that allocate additional computation on test samples during online inference phase. Self-Refine generates an initial output and then provide verbal feedback for subsequent revisions on the same test sample.
> > Similarly, Reflexion incorporates an external feedback signal for reflection and a new attempt during testing on a single sample. A key characteristic of such methods is their focus on iterative, within-sample improvement for a single test sample during online inference. In contrast, Training-Free GRPO optimizes the experiences on a separate training set without accessing any test samples during offline learning, while its online inference remains simple prompting or ReAct using the learned static experiences without any iterative refinement.
> >
> > **References**
> >
> > [1] Leslie Pack Kaelbling, Michael L Littman, and Andrew W Moore. Reinforcement learning: A survey. Journal of artificial intelligence research, 4:237–285, 1996.
> >
> > [2] John Gittins, Kevin Glazebrook, and Richard Weber. Multi-armed bandit allocation indices. John Wiley & Sons, 2011.
> >
> > [3] Kai Arulkumaran, Marc Peter Deisenroth, Miles Brundage, and Anil Anthony Bharath. Deep reinforcement learning: A brief survey. IEEE signal processing magazine, 34(6):26–38, 2017.
> >
> > [4] Qingxiu Dong, Lei Li, Damai Dai, Ce Zheng, Jingyuan Ma, Rui Li, Heming Xia, Jingjing Xu, Zhiyong Wu, Baobao Chang, et al. A survey on in-context learning. In Proceedings of the 2024 conference on empirical methods in natural language processing, pp. 1107–1128, 2024.
> >
> > [5] Qiyuan Zhang, Fuyuan Lyu, Zexu Sun, Lei Wang, Weixu Zhang, Wenyue Hua, Haolun Wu, Zhihan Guo, Yufei Wang, Niklas Muennighoff, et al. A survey on test-time scaling in large language models: What, how, where, and how well? arXiv preprint arXiv:2503.24235, 2025a.

---

### Official Review · Reviewer_YnKW · 2025-10-28

**Soundness:** 3
**Presentation:** 2
**Contribution:** 3
**Rating:** 6
**Confidence:** 3

**Summary:**

The paper proposes Training-free GRPO, an iterative method that builds an experience library by comparing outputs of a base model against each other. Instead of updating model parameters, it refines behavior through accumulated experiential knowledge, offering a lightweight alternative to RL-based fine-tuning. The approach is well-motivated and demonstrates how iterative prompt optimization can achieve competitive results with lower computational cost.

**Strengths:**

The paper explores the idea of iterative prompt optimization as an alternative to RL-based fine-tuning, aiming to reduce the need for costly online samples. The motivation is clear, and the comparison between parameter-space and context-space optimization is conceptually interesting. The experimental setup spans multiple domains and shows consistent improvements across tasks, demonstrating the method’s practicality. The overall structure and writing make the paper easy to follow, although some implementation details are missing.

**Weaknesses:**

### Major comments:

- The method appears heavily dependent on the underlying model’s inherent reasoning strength. For instance, results on Qwen2.5-32B-Instruct are missing in Table 1 but appear in Table 4 (WebWalkerQA), where they show minimal improvement. While the authors briefly acknowledge this at the end of Section 3, it should be emphasized further and discussed in more depth.

- From Table 2, access to ground-truth results seems quite important for the improvements. It would benefit readers to explicitly explain how the ground truth is integrated into the offline dataset in Section 2.

- Table 5 can be misleading: since the base models differ, cross-model comparisons are not meaningful. The relevant comparison should be across domains rather than models. Moreover, Training-free GRPO leverages specialized experience libraries, making the claim of “cross-domain generalization” less convincing. A more rigorous evaluation (e.g., testing experience libraries trained on one domain and applied to another) is needed to support such claims.

### Minor comments:

- Since the optimization happens on the prompt level rather than the policy (usually refered to the base model) itself, the name “GRPO” may not be the most precise.

- How does the base model retrieve relevant experiences from the library? In Figure 6 it appears that the full library is appended to the prompt, whereas in Figure 11 each agent trajectory uses specific experiences. Clarifying this mechanism in Section 2 would make the method much clearer.

- Related work is missing prior studies on offline experience or guideline construction, such as:

[1] Fu et al., AutoGuide: Automated Generation and Selection of Context-Aware Guidelines for Large Language Model Agents;

[2] Wang et al., Agent Workflow Memory.

- It’s reasonable that ablations on WebWalkerQA are done on a subset of tasks, but it is unclear why they are performed after only two epochs of experience optimization. This choice should be justified.

**Questions:**

- When generating $A_{\text{text}_i}$, does the LLM have access to other summaries in the group? From line 194 it seems not, but if so, how can it compute a relative semantic advantage?

- How are “self-generated experiences” produced. Do they follow the same group-comparison pipeline but skip the experience refinement stage?

- In Figure 4, AIME24 and AIME25 show opposite trends in Pass@32 performance across steps. What might explain this behavior?

- What happens if the optimization runs for more than three epochs (e.g., 5 or 10)? Would the experience library become more specialized, or would overfitting occur?

---

> ### Author Response · Authors · 2025-12-03
>
> We appreciate your thoughtful feedback. We would like to clarify the points you raised as follows:
>
> ## For Major Comments 1
>
> In the revised Section 3.2, we further explicitly acknowledge that the effectiveness of context-based RL optimization is dependent on the underlying model’s intrinsic reasoning and introspection capabilities, indicating that certain model capability is a prerequisite for effectively applying Training-Free GRPO
>
> ## For Major Comments 2
>
> As shown in the revised Section 3.2, including ground truth benefits the performance, since the it is more easy to contrast succesful and failed trajectories for distilling better experiences. As suggested, the revised Section 2 mentions how to use the ground truth during offline learning. Specifically, the reward model determines whether the answer $o_i$ matches the ground truth $y$, producing the scalar reward  $r_i=\mathcal{R}(o_i, y)$. Moreover, such groud truths are usually associated with well-established datasets. In our experiments, both DAPO-Math-17k and AFM-web-RL datasets contains the final answer to the problem without step-by-step solution.
>
> ## For Major Comments 3
>
> We agree with your suggestion of meaningful comparison. In Section 3.2 of the revised manuscript, we compare Training-Free GRPO against multiple vanilla RL methods on identical LLMs, including both Qwen2.5-32B-Instruct and DeepSeek-V3.1-Terminus. As expected, parameter tuning offers a larger search space and typically yields higher performance ceilings than optimizaing prompts. However, parameter-based specialization is susceptible to catastrophic forgetting, narrowing the model’s capabilities to the training domain at the expense of generalizability. In contrast, Training-Free GRPO circumvents this issue by maintaining a single, general-purpose frozen LLM. It allows for flexible domain switching simply by plugging the corresponding learned experiences.
>
> We also agree with your more rigorous test of transferablity, and have accordingly revised the manuscript in Section 3.2. As expected, Training-Free GRPO remains much more robust than RL trained models. Notably, different from the higher costs of deploying multiple specialized RL trained models, Training-Free GRPO will never use experiences from opposite domains in practice, since it runs on a single LLM and requires no extra cost for deploying on all domains.
>
> ## For Minor Comments 1
>
> As discussed in the revised Section 4, Training-Free GRPO is a RL method during offline learning. Following the fundamental definition [1], RL is a paradigm of learning through trial-and-error to maximize rewards by optimizing a policy. A policy is defined as a mapping from states to actions [1], which could be simple look-up tables [2]. Also, RL optimization could be gradient-free methods [3].
> In the Section 1 of the revised manuscript, we clarify that Training-Free GRPO satisfies this fundamental definition of RL. It instantiates the RL policy as the union of a frozen LLM and its variable context, as illustrated in the revised Figure 1(a). Also, during its multi-epoch learning process, the policy context is optimized through trial-and-error to maximize rewards.
>
> Furthermore, we use the name of Training-Free GRPO since it actually adopts the exact process of group relative policy optimization, producing a group of rollouts and optimize the policy according to the relative quality of rollouts. Comparing with GRPO (Group Relative Policy Optimization), our method only differs on the definition of policy and the process of optimization as discussed above.
>
> ## For Minor Comments 2
>
> As suggested, we have revised Section 2 and clarify that we directly inject all the current experiences $\mathcal{E}$ into the context.
>
> ## For Minor Comments 3
>
> As suggested, we have included these studies and revised Section 4 accordingly. AutoGuide [4] generates context-aware guidelines from offline data in a one-pass manner. Agent Workflow Memory (AWM) [5] induces workflows exclusively from successful trajectories and integrates into memory.
>
> However, our approach is distinguished by two key factors:
> - Multi-Round Iterative Optimization: While they typically extract knowledge in a single pass during offline, Training-Free GRPO treats experiential knowledge as an RL policy, employing multi-epoch learning to iteratively optimize it.
> - Contrastive Experience Distillation: They derive insights solely from single trajectories, but Training-Free GRPO contrasts multiple successful and failed trajectories for the same query, extracting more robust experiences as validated in Section 3.3.
>
> ## For Minor Comments 4
>
> As pointed out, we have revised the ablation experiments in Section 3.3, using consistent 3 epoches and reporting new results.

---

> > ### Author Response · Authors · 2025-12-03
> >
> > ## For Question 1
> >
> > When generating semantic group advantage $A_{text_i}$, the LLM can access to other summaries in the group. In the revised Section 2, we have formalized the process and explicitly stated that "With such summaries $\mathbf{s}=\{s_1, s_2, \dots, s_G\}$, the LLM M then extracts the $A_{text_i}=M({p_{\text{adv}}}, q, i, \mathbf{s}, y, \mathbf{r})$ for each output $o_i$, where $p_{\text{adv}}$ is the prompt template for advantage generation."
> >
> > ## For Question 2
> >
> > The self-generated experiences are produced by directly prompting DeepSeek-V3.1-Terminus to generate synthesis experiments for matching the format and quantity learned from Training-Free GRPO.
> > It is not related to any group-comparison pipeline. In the revised Section 3.3, we have clarified this explicitly.
> >
> > ## For Question 3
> >
> > In this paper, we focus primarily on Mean@K, which offers greater robustness by averaging across K independent runs. In contrast, Pass@K measures whether at least one solution succeeds in K attempts, which can exhibit higher variability, especially when the model's performance is near the threshold of solving a problem.
> >
> > ## For Question 4
> >
> > As suggested, we try to extend the optimization epoch to $5$, resulting in a performance decline at the $5^{th}$ learning epoch. This indicates that accumulating excessive experiences may eventually saturate the context or introduce noise. And in practice, it is necessary to halt optimization at the appropriate epoch, such as epoch $3$ in our experiments.
> >
> > **References**
> >
> > [1] Leslie Pack Kaelbling, Michael L Littman, and Andrew W Moore. Reinforcement learning: A survey. Journal of artificial intelligence research, 4:237–285, 1996.
> >
> > [2] John Gittins, Kevin Glazebrook, and Richard Weber. Multi-armed bandit allocation indices. John Wiley & Sons, 2011.
> >
> > [3] Kai Arulkumaran, Marc Peter Deisenroth, Miles Brundage, and Anil Anthony Bharath. Deep reinforcement learning: A brief survey. IEEE signal processing magazine, 34(6):26–38, 2017.
> >
> > [4] Yao Fu, Dong-Ki Kim, Jaekyeom Kim, Sungryull Sohn, Lajanugen Logeswaran, Kyunghoon Bae, and Honglak Lee. Autoguide: Automated generation and selection of context-aware guidelines for large language model agents. Advances in Neural Information Processing Systems, 37: 119919–119948, 2024.
> >
> > [5] Zora Zhiruo Wang, Jiayuan Mao, Daniel Fried, and Graham Neubig. Agent workflow memory. arXiv preprint arXiv:2409.07429, 2024d.

---

### Official Review · Reviewer_H5m6 · 2025-10-31

**Soundness:** 2
**Presentation:** 3
**Contribution:** 2
**Rating:** 4
**Confidence:** 5

**Summary:**

This paper introduces Training-free Group Relative Policy Optimization (Training-free GRPO), a method for enhancing LLM agent performance without updating model parameters. The approach leverages group-based rollouts to distill a natural language "semantic advantage" from successful and failed attempts, which is then used to iteratively build and refine an external "experience library". This library is provided as context (a "token prior") to a frozen LLM, effectively steering its output distribution towards higher-reward behaviors in a data- and compute-efficient manner.

**Strengths:**

1. The central idea of shifting policy optimization from the parameter space to the context space is a practical concept. It presents a compelling alternative to traditional fine-tuning, directly addressing critical challenges such as prohibitive computational costs, data scarcity, and potential catastrophic forgetting associated with parameter updates.
2. The paper demonstrates the effectiveness of the proposed method with good empirical results on challenging and well-recognized benchmarks for mathematical reasoning (AIME24/25) and web navigation (WebWalkerQA). The application to a capable frozen model (DeepSeek-V3.1) shows significant performance gains, highlighting the practical utility of the approach.
3. The concept is explained with a helpful analogy to vanilla GRPO, and the inclusion of a concrete example in Figure 3 effectively illustrates the core mechanism of generating and applying "semantic advantage". This aids in understanding the intuition behind the method.

**Weaknesses:**

1. The experimental evaluation suffers from a lack of direct, controlled comparisons, making it difficult to isolate the true contribution of the proposed method. The primary results in Table 1 compare Training-free GRPO on the powerful DeepSeek-V3.1 model against RL baselines fine-tuned on the weaker Qwen2.5-32B model. To make a convincing claim, the paper should have included results for Training-free GRPO on Qwen2.5-32B and, more importantly, a traditional GRPO fine-tuning baseline on DeepSeek-V3.1. Without these crucial control experiments, the observed performance gap could be attributed more to the superior base model than to the merits of the training-free optimization technique.
2. The conceptual framing and terminology used in the paper are potentially misleading. The name "Training-free GRPO" is an oxymoron, as GRPO is fundamentally a training algorithm that updates policy parameters, which this method explicitly avoids. Furthermore, classifying the method as "RL" in Table 1 is questionable. Since the model parameters are frozen and improvement comes from engineering the context, the method is mechanistically closer to iterative prompt optimization or in-context learning rather than reinforcement learning in the traditional sense. This framing creates confusion and potentially overstates the connection to established RL algorithms.
3. The cross-domain transfer analysis in Section 4.1 is not designed as a fair comparison. The experiment shows that specialized fine-tuned models (ReTool, MiroThinker) perform poorly when transferred to a new domain, which is a well-known limitation. However, the Training-free GRPO method is evaluated by providing it with a domain-specific experience library for each task. A more rigorous test of transferability would involve using the experiences learned from the math domain to evaluate performance on the web search domain, which would properly assess how the learned knowledge generalizes, rather than just showing the flexibility of swapping context.
4. The ablation studies could be more comprehensive and the methodology more clearly defined. The paper states that semantic advantage is generated only for groups with "clear winners and losers," but this critical condition is not formally defined. The sensitivity to key hyperparameters, such as the group size $G$, is not explored. Furthermore, the term "reward model" used in Figure 2 is ambiguous; for verifiable tasks like math, this is typically a deterministic reward function, and using the word "model" incorrectly implies a learned, parameterized critic.
5. The important negative result on the weaker QwQ-32B model, where the method underperforms its own ReAct baseline, is a significant finding that is not sufficiently discussed as a core limitation. This suggests that the effectiveness of Training-free GRPO is highly dependent on the advanced reasoning and introspection capabilities of the underlying base model, which should be stated more explicitly as a prerequisite for the method's success.

**Questions:**

1. The paper states that semantic advantages are generated only for groups with "clear winners and losers" (line 189). Could you please precisely define this condition? Is it based on the variance of rewards, or is there another mechanism at play? How does your method handle the common scenario where all trajectories in a group fail and receive an identical low reward?
2. Could you justify the experimental design choice of not including direct comparisons on the same base model? Specifically, why was a traditional fine-tuning method like GRPO not applied to DeepSeek-V3.1, and why were results for Training-free GRPO on Qwen2.5-32B not reported? Is the observed performance advantage primarily due to the strength of the base model?
3. Could you clarify the conceptual framing of your method? Specifically, please justify the name "Training-free GRPO" given that no policy parameters are updated, and explain why it is classified as "RL" rather than an advanced form of prompt engineering, especially since other iterative methods like ReAct are categorized as "Prompt". Additionally, could you confirm whether the "reward model" is a fixed, deterministic function rather than a learned model?
4. Regarding the cross-domain analysis, would you consider running a more stringent experiment where the experience library learned on the math tasks is directly applied to the web navigation tasks without modification? This would provide a much clearer assessment of the generalizability of the learned "experiential knowledge".
5. The reference section contains several critical errors (e.g., the DeepSeek AI reference cites dates in 2025, the Li et al., 2025a and 2025b citations appear identical). We strongly urge the authors to perform a thorough revision of the bibliography. Could you confirm that these will be corrected?
6. Please describe the differences and connections between your proposed method and the ReasonFlux series of papers.
https://arxiv.org/pdf/2502.06772, https://arxiv.org/pdf/2506.18896, https://arxiv.org/pdf/2506.03136

---

> ### Author Response · Authors · 2025-12-03
>
> We appreciate your thoughtful feedback. We would like to clarify the points you raised as follows:
>
> ## For Weakness 1 and Question 2
>
> In this paper, we focus on practical scenarios of adapting LLMs to specialized tasks. Regardless of model size, the central questions we aim to answer are: Which approach offers the best cost-efffectiveness for real-world deployment? As a strategic decision for our own business, which one promises a better return on investment? As confirmed in Figure 1(b)-(c) of the revised manuscript, comparing to the RL-trained 32B LLM, Training-Free GRPO achieves higher performance with much lower learning costs.
>
> We also agree with your suggestion of control experiments on the indentical LLMs. In Section 3.2 of the revised manuscript, we compare Training-Free GRPO against vanilla RL methods on identical LLMs, including both Qwen2.5-32B-Instruct and DeepSeek-V3.1-Terminus. As expected, parameter tuning offers a larger search space and typically yields higher performance ceilings than optimizaing prompts. However, parameter-based specialization is susceptible to catastrophic forgetting, narrowing the model’s capabilities to the training domain at the expense of generalizability. In contrast, Training-Free GRPO circumvents this issue by maintaining a single, general-purpose frozen LLM. It allows for flexible domain switching simply by plugging the corresponding learned experiences.
>
> ## For Weakness 2 and Question 3
>
> As discussed in the revised Section 4, Training-Free GRPO is a RL method during offline learning, and it is also an ICL method during online inference. But it is not a Test-Time Scaling method.
> - Reinforcement Learning (RL): Following the fundamental definition [1], RL is a paradigm of learning through trial-and-error to maximize rewards by optimizing a policy. A policy is defined as a mapping from states to actions [1], which could be simple look-up tables [2]. Also, RL optimization could be gradient-free methods [3].
> In the Section 1 of the revised manuscript, we clarify that Training-Free GRPO satisfies this fundamental definition of RL. It instantiates the RL policy as the union of a frozen LLM and its variable context, as illustrated in the revised Figure 1(a). Also, during its multi-epoch learning process, the policy context is optimized through trial-and-error to maximize rewards.
> - In-Context Learning (ICL): Training-Free GRPO include the learned experiential knowledge in the context during online inference, satisfying the broader idea of ICL, i.e., learn from analogy [4].
> - Distinction from Test-Time Scaling (TTS): TTS allocates additional computation for each test sample (e.g., Tree of Thoughts, Self-Refine) during online inference or testing [5]. In contrast, Training-Free GRPO optimizes experiences on a separate training set without access to any test samples, as shown in the revised Figure 1(a). Once these optimized experiences are incorporated into the context, online inference proceeds with simple prompting or ReAct, requiring no extra computation at test time. In essence, the method differs from standard prompting or ReAct only through its use of learned static context. and does not require any additional computation during testing. Thus, Training-Free GRPO is orthogonal to TTS methods and can be readily combined with any TTS strategy.
>
> Furthermore, we use the name of Training-Free GRPO since it actually adopts the exact process of group relative policy optimization, producing a group of rollouts and optimize the policy according to the relative quality of rollouts. Comparing with GRPO (Group Relative Policy Optimization), our method only differs on the definition of policy and the process of optimization as discussed above.
>
> ## For Weakness 3 and Question 4
>
> We agree with your more rigorous test of transferablity, and have accordingly revised the manuscript in Section 3.2. As expected, Training-Free GRPO remains much more robust than RL trained models. Notably, different from the higher costs of deploying multiple specialized RL trained models, Training-Free GRPO will never use experiences from different domains in practice, since it runs on a single LLM and requires no extra cost for deploying on all domains.

---

> > ### Author Response · Authors · 2025-12-03
> >
> > ## For Weakness 4 and 5, Questions 1, 3 and 5
> >
> > We have formally presented the suggested contents in the revised manuscript as follows:
> > - In the revised Section 2, for the conditioin of generating semantic advantage, the unclear statement of "clear winners and losers" has been replaced with "for groups with $\text{std}(\mathbf{r})=0$, $A_{\text{text}_i}$ will not be generated". And when all trajectories receive an identical reward, there is no advantage used for optimization, which is exactly the same solution as vanilla GRPO.
> > - In the revised Section 3.3, we also include the ablation of varying group size. Results show that larger group size tends to achieve better performance.
> > - In the revised Section 2, we have clarified the definition of reward model, which could be a rule-based function or an LLM judging whether $o_i$ matches the ground truth $y$, producing the scalar reward $r_i=\mathcal{R}(o_i, y)$.
> > - In the revised Section 3.2, we have explicitly stated that "the effectiveness of context-based RL optimization is dependent on the underlying model’s intrinsic reasoning and introspection capabilities, indicating that certain model capability is a prerequisite for effectively applying Training-Free GRPO."
> > - We have remove the duplicated bib item as pointed out.
> >
> > ## For Question 6
> >
> > As presented in Section 4 of the revised manuscript, ReasonFlux [6] and its variants [7,8] construct thought templates by analyzing the reasoning behind individual solutions in a single pass.
> >
> > However, our approach is distinguished by two key factors:
> > - Multi-Round Iterative Optimization: While ReasonFlux extracts templates in a single pass during offline, Training-Free GRPO treats experiential knowledge as an RL policy, employing multi-epoch learning to iteratively optimize it.
> > - Contrastive Experience Distillation: ReasonFlux derives templates solely from single successful trajectories, but Training-Free GRPO contrasts multiple successful and failed trajectories for the same query, extracting more robust experiences as validated in Section 3.3.
> >
> > **References**
> >
> > [1] Leslie Pack Kaelbling, Michael L Littman, and Andrew W Moore. Reinforcement learning: A survey. Journal of artificial intelligence research, 4:237–285, 1996.
> >
> > [2] John Gittins, Kevin Glazebrook, and Richard Weber. Multi-armed bandit allocation indices. John Wiley & Sons, 2011.
> >
> > [3] Kai Arulkumaran, Marc Peter Deisenroth, Miles Brundage, and Anil Anthony Bharath. Deep reinforcement learning: A brief survey. IEEE signal processing magazine, 34(6):26–38, 2017.
> >
> > [4] Qingxiu Dong, Lei Li, Damai Dai, Ce Zheng, Jingyuan Ma, Rui Li, Heming Xia, Jingjing Xu, Zhiyong Wu, Baobao Chang, et al. A survey on in-context learning. In Proceedings of the 2024 conference on empirical methods in natural language processing, pp. 1107–1128, 2024.
> >
> > [5] Qiyuan Zhang, Fuyuan Lyu, Zexu Sun, Lei Wang, Weixu Zhang, Wenyue Hua, Haolun Wu, Zhihan Guo, Yufei Wang, Niklas Muennighoff, et al. A survey on test-time scaling in large language models: What, how, where, and how well? arXiv preprint arXiv:2503.24235, 2025a.
> >
> > [6] Ling Yang, Zhaochen Yu, Bin Cui, and Mengdi Wang. Reasonflux: Hierarchical llm reasoning via scaling thought templates. arXiv preprint arXiv:2502.06772, 2025b.
> >
> > [7] Jiaru Zou, Ling Yang, Jingwen Gu, Jiahao Qiu, Ke Shen, Jingrui He, and Mengdi Wang. Reasonflux-prm: Trajectory-aware prms for long chain-of-thought reasoning in llms. arXiv preprint arXiv:2506.18896, 2025.
> >
> > [8] Yinjie Wang, Ling Yang, Ye Tian, Ke Shen, and Mengdi Wang. Co-evolving llm coder and unit tester via reinforcement learning. arXiv preprint arXiv:2506.03136, 2025.

---

### Official Review · Reviewer_xdfL · 2025-11-02

**Soundness:** 2
**Presentation:** 3
**Contribution:** 2
**Rating:** 2
**Confidence:** 4

**Summary:**

This work introduces training-free GRPO a method that mimics the traditional gradient updates of GRPO with contextual aggregation of trials.

**Strengths:**

The paper presents a decent idea with their aggregation mechanism for rationales and incorporates a self judging mechanism with the same LLM that seems to be fairly simple and efficient to implement. The presentation and plots in the paper a nice making it easy to read and understand. Overall I think the papers overall presentation and method are sound.

**Weaknesses:**

With the strengths being said, I think the paper has significant weaknesses that I think prevent me from recommending its acceptance. First, the marketing of the paper I have fundamental problems with, it is written in a way that makes it seem like through simple pseudo-prompting one can learn new things similar to what could be learnt through RL. I believe this to be simply not true, as if there are problems the model cannot solve (i.e the answer is not in its support) no amount of reasonable prompting (except maybe leaking the answer) should be able to obtain this sample. Now test-time scaling/online ICL does work, but I am reluctant to draw an equivalence between it and RL.

With that said,  I think the paper suffers from further technical weaknesses. It is a bit weird to be that training-free GRPO is implemented for Deepseek V1 but RL is not implemented for the same model, only for Qwen 32b. Since this is ultimately a psu-prompting work I would expect more complex agentic pipelines being compared, I am unfamiliar with that side of the literature but something akin to [1].

Multiple random seeds and multiple models are also missing. Overall I think the work lacks significant polishing.


[1] https://arxiv.org/abs/2509.26626

**Questions:**

See weaknesses

---

> ### Author Response · Authors · 2025-12-03
>
> We appreciate your thoughtful feedback. We would like to clarify the points you raised as follows:
>
> ## Comparing Context-based and Parameter-based Optimization
> In Section 3.2 of the revised manuscript, we compare Training-Free GRPO against vanilla RL methods on identical LLMs, including both Qwen2.5-32B-Instruct and DeepSeek-V3.1-Terminus.
> - We agree that parameter tuning offers a larger search space and typically yields higher performance ceilings than optimizaing prompts. However, parameter-based specialization is susceptible to catastrophic forgetting, narrowing the model’s capabilities to the training domain at the expense of generalizability. In contrast, Training-Free GRPO circumvents this issue by maintaining a single, general-purpose frozen LLM. It allows for flexible domain switching simply by plugging the corresponding learned experiences.
> - As shown in Figure 1(b)-(c) in the revised manuscript, Training-Free GRPO does effectively interpolate the Pareto frontier, offering a significantly lower learning cost while delivering meaningful performance improvements.
>
> ## Discusing Problem Solvability during Learning Process
> As confirmed by experiments, Training-Free GRPO does achieve overall improvments without leaking answer.
> - By increasing the temperature, vanilla RL methods like GRPO rely entirely on the model's own ability to generate a correct trajectory during the rollout phase. If a problem is truly "outside the support" of the model, the advantage in GRPO would be 0. But for those partially correct problems, it can distill the model's latent capabilities illustrated by Pass@K into consistent Mean@K performance [1].
> - Similarly, our Training-Free GRPO increases the temperature, encouraging the LLM to generate a group of trials and distill experiences by contrasting successful and falied trajectories. This can uncover valid reasoning pattern that were previously low-probability, and Training-Free GRPO presents them in the context for future inference, achieving the similar effect of distilling latent capabilities illustrated by Pass@K into consistent Mean@K performance.
>
> ## Categorization of Our Method
> As discussed in the revised Section 4, Training-Free GRPO is a RL method during offline learning, and it is also an ICL method during online inference. But it is not a Test-Time Scaling method.
> - Reinforcement Learning (RL): Following the fundamental definition [2], RL is a paradigm of learning through trial-and-error to maximize rewards by optimizing a policy. A policy is defined as a mapping from states to actions [2], which could be simple look-up tables [3]. Also, RL optimization could be gradient-free methods [4].
> In the Section 1 of the revised manuscript, we clarify that Training-Free GRPO satisfies this fundamental definition of RL. It instantiates the RL policy as the union of a frozen LLM and its variable context, as illustrated in the revised Figure 1(a). Also, during its multi-epoch learning process, the policy context is optimized through trial-and-error to maximize rewards.
> - In-Context Learning (ICL): Training-Free GRPO include the learned experiential knowledge in the context during online inference, satisfying the broader idea of ICL, i.e., learn from analogy [5].
> - Distinction from Test-Time Scaling (TTS): TTS allocates additional computation for each test sample (e.g., Tree of Thoughts, Self-Refine) during online inference or testing [6]. In contrast, Training-Free GRPO optimizes experiences on a separate training set without access to any test samples, as shown in the revised Figure 1(a). Once these optimized experiences are incorporated into the context, online inference proceeds with simple prompting or ReAct, requiring no extra computation at test time. In essence, the method differs from standard prompting or ReAct only through its use of learned static context. and does not require any additional computation during testing. Thus, Training-Free GRPO is orthogonal to TTS methods and can be readily combined with any TTS strategy.
>
> ## Comparing with Agentic Pipelines
> Agentic pipelines are categorized as Test-Time Scaling (TTS) methods in recent survey [6], since it allocates additional computation for each test sample during online inference. As discussed above, Training-Free GRPO is orthogonal to TTS methods, thus it is improper for direct comparison. And in the future work, we will explore combining Training-Free GRPO with TTS methods during online inference, e.g., injecting the offline learned experiences into the context of complex agentic pipelines.

---

> > ### Author Response · Authors · 2025-12-03
> >
> > **References**
> >
> > [1] Yang Yue, Zhiqi Chen, Rui Lu, Andrew Zhao, Zhaokai Wang, Yang Yue, Shiji Song, Gao Huang. Does reinforcement learning really incentivize reasoning capacity in llms beyond the base model?. arXiv preprint arXiv:2504.13837 (2025).
> >
> > [2] Leslie Pack Kaelbling, Michael L Littman, and Andrew W Moore. Reinforcement learning: A survey. Journal of artificial intelligence research, 4:237–285, 1996.
> >
> > [3] John Gittins, Kevin Glazebrook, and Richard Weber. Multi-armed bandit allocation indices. John Wiley & Sons, 2011.
> >
> > [4] Kai Arulkumaran, Marc Peter Deisenroth, Miles Brundage, and Anil Anthony Bharath. Deep reinforcement learning: A brief survey. IEEE signal processing magazine, 34(6):26–38, 2017.
> >
> > [5] Qingxiu Dong, Lei Li, Damai Dai, Ce Zheng, Jingyuan Ma, Rui Li, Heming Xia, Jingjing Xu, Zhiyong Wu, Baobao Chang, et al. A survey on in-context learning. In Proceedings of the 2024 conference on empirical methods in natural language processing, pp. 1107–1128, 2024.
> >
> > [6] Qiyuan Zhang, Fuyuan Lyu, Zexu Sun, Lei Wang, Weixu Zhang, Wenyue Hua, Haolun Wu, Zhihan Guo, Yufei Wang, Niklas Muennighoff, et al. A survey on test-time scaling in large language models: What, how, where, and how well? arXiv preprint arXiv:2503.24235, 2025a.

---

### Author Response · Authors · 2025-12-03

We sincerely thank all the reviewers for their constructive comments, which have significantly helped in polishing our manuscript. Most of the concerns center around four aspects: 1) The conceptual framing and categorization of the method; 2) The fairness of experimental comparisons; 3) Technical definitions and ablation details; and 4) Discussions on related work and limitations.

We address the above concerns by:
- For the categorization of the method, we clarified in the revised Section 1 and 4 that Training-Free GRPO satisfies the fundamental definition of Reinforcement Learning, i.e., optimizing a policy via trial-and-error to maximize rewards, where the policy is the union of a frozen LLM and its variable context. We explicitly differentiate our method from Test-Time Scaling (TTS), as our method requires no additional computation during the online inference phase, making it orthogonal to TTS strategies.
- For experimental fairness, we have added controlled comparisons on identical base models in the revised Section 3.2. We now compare Training-Free GRPO against vanilla RL methods on both Qwen2.5-32B-Instruct and DeepSeek-V3.1-Terminus. While parameter tuning offers a higher ceiling, we demonstrate that our method avoids catastrophic forgetting and offers cost-effectiveness on the Pareto frontier.
- For technical details and definitions, we have clarified the condition for generating semantic advantages and the definition of reward model. We also added ablation studies on group size in Section 3.3 and explicitly acknowledged that high intrinsic model capability is a prerequisite for our method, explaining the performance gap on smaller models.
- For related work and writing, we have refined the discussion in Section 4 to distinguish our work from similar methods. We have also refined the writing throughout the manuscript.

All the comments are addressed in the rebuttal, and we have performed modifications on the manuscript accordingly.

---

### Meta-Review · Area_Chair_Agoz · 2026-01-04

**Summary:**

The reviewers raised the following major concerns. First, many reviewers pointed out weaknesses in experimental design, particularly (i) unfair comparisons arising from implementing different methods with different underlying models, and (ii) insufficient empirical evaluation in terms of baselines and benchmarks. Second, reviewers expressed concerns that the limitations of the proposed method were not clearly stated, namely, that its performance heavily depends on the quality of the underlying LLM it leverages, which restricts the broader applicability of the proposed approach. Finally, there were concerns regarding paper presentation.

**Reviewer Concerns:**

The rebuttal and revised manuscript addressed concerns related to paper presentation. The authors also clearly acknowledged the limitation of the proposed method.

In response to the first concern above, the authors provided empirical comparisons of different methods with the same underlying model. When implementing with the same LLM, GRPO shows a clear advantage over the proposed training-free GRPO method. While the proposed method is computationally more efficient, the substantial performance gap raises new concerns regarding the need for more comprehensive evaluations against additional baselines. Results against other baselines or on additional benchmarks were not provided during rebuttal.

**Reviewer Scores:**

The added experimental results comparing different methods with the same underlying model reveal a clear performance gap between GRPO and the proposed method. While the proposed method is computationally more efficient, the substantial performance gap raises concerns regarding the need for more comprehensive evaluations against additional baselines. As a result, I believe the reviewers are unlikely to increase their scores.

---

### Decision · Program_Chairs · 2026-01-26

Reject